# Identification of key yeast species and microbe–microbe interactions impacting larval growth of *Drosophila* in the wild

Ayumi Mure[1], Yuki Sugiura[2], Rae Maeda[2], Kohei Honda[1], Nozomu Sakurai[3], Yuuki Takahashi[1], Masayoshi Watada[4], Toshihiko Katoh[1], Aina Gotoh[1], Yasuhiro Gotoh[5], Itsuki Taniguchi[5], Keiji Nakamura[5], Tetsuya Hayashi[5], Takane Katayama[1], Tadashi Uemura[1,6,7], Yukako Hattori[1,6,8]*

[1]Graduate School of Biostudies, Kyoto University, Kyoto, Japan; [2]Center for Cancer Immunotherapy and Immunobiology, Kyoto University, Kyoto, Japan; [3]National Institute of Genetics, Mishima, Japan; [4]Graduate School of Science and Engineering, Ehime University, Matsuyama, Japan; [5]Graduate School of Medical Sciences, Kyushu University, Fukuoka, Japan; [6]Center for Living Systems Information Science, Kyoto University, Kyoto, Japan; [7]AMED-CREST, Tokyo, Japan; [8]JST FOREST, Tokyo, Japan

*For correspondence:
yhattori@lif.kyoto-u.ac.jp

Competing interest: The authors declare that no competing interests exist.

**Abstract** Microbiota consisting of various fungi and bacteria have a significant impact on the physiological functions of the host. However, it is unclear which species are essential to this impact and how they affect the host. This study analyzed and isolated microbes from natural food sources of *Drosophila* larvae, and investigated their functions. *Hanseniaspora uvarum* is the predominant yeast responsible for larval growth in the earlier stage of fermentation. As fermentation progresses, *Acetobacter orientalis* emerges as the key bacterium responsible for larval growth, although yeasts and lactic acid bacteria must coexist along with the bacterium to stabilize this host–bacterial association. By providing nutrients to the larvae in an accessible form, the microbiota contributes to the upregulation of various genes that function in larval cell growth and metabolism. Thus, this study elucidates the key microbial species that support animal growth under microbial transition.

## eLife assessment

This is an **important** study that addresses a significant question in microbiome research. The authors provide **convincing** evidence that certain bacterial groups within the fly microbiome have critical functions for host development. Additionally, dietary aspects such as microbial community progression in a natural food source are integrated into their host-microbe interaction analyses.

## Introduction

In nature, animals live in association with a diverse community of microorganisms. These associated microbes, especially fungi and bacteria, perform a range of beneficial functions for their host, such as nutrient provision and immune modulation (*Ikeda-Ohtsubo et al., 2018*; *Zheng et al., 2020*). Some of these host–microbe associations are facultative and dispensable, while others are more important or even essential for host growth or survival under specific circumstances. Compared to symbioses between a limited number of specific species (*Shigenobu and Wilson, 2011*; *Su et al., 2022*), however, those that encompass a larger number of species are intricate and analytically challenging. Additionally, despite a growing number of reports documenting the presence of fungi in the mammalian intestine, including humans, our comprehension of their roles is still limited, with the

exception of a handful of pathogenic fungi (*Hallen-Adams and Suhr, 2017*; *Pérez, 2021*). Moreover, the composition of these complex microbiota tends to change over time (*Moya and Ferrer, 2016*; *Qiu et al., 2021*). Therefore, we focused on how the host sustains its life processes in such unstable environments.

*Drosophila melanogaster* has made remarkable contributions to our comprehension of the regulatory mechanisms governing development, growth, and metabolism, which are highly conserved among animals. More recently, the fly has emerged as a valuable model for investigating animal-associated microbes. The microbiota associated with *Drosophila* comprise relatively few species, most of which can be cultured aerobically (*Chandler et al., 2011*; *Chandler et al., 2012*; *Grenier and Leulier, 2020*; *Lee and Brey, 2013*). Furthermore, germ-free (GF) or gnotobiotic animals can be easily prepared (*Ludington and Ja, 2020*). The aforementioned advantages of analysis enabled a thorough exploration of the role of *Drosophila*-associated microbes. Notably, lactic acid bacteria (LAB) and acetic acid bacteria (AAB) demonstrate the ability to enhance larval growth under nutrient scarcity (*Shin et al., 2011*; *Storelli et al., 2011*). However, most researchers have examined only a single species or a limited number of bacteria. Additionally, most previous studies did not investigate the association with yeasts. Possibly, this is because live yeasts are either absent or present in very small quantities, due to antifungal agents added to the *Drosophila* laboratory foods. Instead of live yeasts, the foods typically contain heat-killed budding yeast *Saccharomyces cerevisiae* as a major nutrient source, but this yeast species rarely coexists with *Drosophila* in the wild (*Hoang et al., 2015*). Consequently, the relationship between *Drosophila* and its associated microorganisms has often been classified as facultative (*Gallo et al., 2022*; *Martino et al., 2018*), and the role played by associated fungi has been largely overlooked.

In the wild, the presence of microbes is crucial for the developmental growth of *Drosophila* larvae that feed on fruit-based food sources. *D. melanogaster* in its natural habitat feeds on fruits fermented by its associated microbes (*Watanabe et al., 2019*). GF larvae cannot grow on fresh fruits alone, while inoculation with certain species of yeasts or bacteria promotes pupariation (*Anagnostou et al., 2010*; *Pais et al., 2018*). These findings suggest that, in the wild, microbes provide vital nutrients for larval growth. While the bacterial roles have been extensively investigated, few reports have focused on yeasts; some studies described interspecies variation of yeasts with regard to their effects on larval growth (*Anagnostou et al., 2010*; *Quan and Eisen, 2018*; *Solomon et al., 2019*), but the underlying mechanisms have not been thoroughly explored. We therefore set out to address the following questions: (1) which species play a central role in food microbiota? (2) what are the essential traits that these species possess? (3) what microbe-derived nutrients are necessary for host growth?

To address these questions, we sampled fermented bananas that had been fed upon by wild *Drosophila* species. We collected the foods at two different timepoints, referred to as 'early-stage' and 'late-stage' foods, and demonstrated a dramatic shift in fungal and bacterial taxonomic compositions during fermentation. Regarding fungi, we observed that yeasts predominated in both stages, but the dominant species changed between the stages. Among bacterial species, Enterobacterales accounted for a large proportion in the early stage, whereas LAB and AAB dominated in the late stage. We then isolated yeast and bacterial strains from the food samples and tested their ability to support larval development on a banana agar. *Hanseniaspora uvarum*, the predominant yeast species in the early stage, was able to support larval growth by itself. In contrast, most of the late-stage microbes tested did not efficiently promote larval growth when inoculated individually. However, we found that when the AAB *Acetobacter orientalis* coexisted with either LAB or late-stage yeast species, it effectively promoted larval growth. Our analyses of larvae under different microbial environments, including transcriptomic analyses of first instars, strongly suggest that *A. orientalis* is potentially able to promote larval growth, although it requires interactions with other late-stage microbes. Finally, we investigated the molecular basis underlying the distinct larval growth-promoting effects among yeast species, including the supportive *H. uvarum* from the early-stage foods and non-supportive *Pichia kluyveri* and *Starmerella bacillaris* from the late-stage foods. Surprisingly, all the yeast species strongly promoted larval growth upon heat killing. This and additional results indicate that all species produce sufficient nutrients for larval development, but larvae cannot utilize those produced by the live non-supportive species. Our metabolomic analysis and metabolite supplementation assay suggest that only the supportive yeast cells can release critical metabolites for larval growth, including branched-chain amino acids (BCAAs), leucine and/or isoleucine. Collectively, our findings detail the key microbial

species, their interactions, and the yeast species-dependent supply of nutrients that contribute to the development of *Drosophila* larvae in nature.

## Results

### The composition of both yeast and bacteria in the food microbiota shifts as fermentation progresses, independently of the presence of larvae

To examine the community structure of the microbiota associated with *Drosophila* larvae in nature, we collected and analyzed larval foods, fermented bananas, that had been fed on by wild larvae (*Figure 1A*, *Supplementary file 1A, B*). Because *D. melanogaster* and its related species are often found near human settlements, we set traps containing freshly peeled and sliced bananas outside of our residences so that wild flies would lay eggs on the foods. A portion of each food sample was collected as 'early-stage' food, while the remaining food was incubated in the laboratory. When the larvae reached the late third instar stage, we collected further fermented 'late-stage' food samples (refer to Materials and methods for experimental definitions of early- or late-stage foods). We performed sequencing of the fungal internal transcribed spacer (ITS) region and the bacterial 16S rRNA region to analyze the composition of fungi and bacteria in individual food samples, respectively (*Figure 1B* and *Figure 1—figure supplement 1*; *Supplementary file 2A, B*).

In most food samples, yeasts consistently accounted for a major proportion of the fungal communities (top of *Figure 1B*; *Figure 1—figure supplement 1A*; *Supplementary file 2A*), as previously reported (*Chandler et al., 2012*). At the family level, the compositions showed dramatic differences between the early and late stages; while Saccharomycodaceae dominated in most of the early-stage foods, Pichiaceae, *Starmerella*, and Saccharomycetaceae became dominant in the late-stage foods. The dominant bacteria also differed between the two stages, with Enterobacterales predominating at the early stage and LAB and AAB predominating at the late stage (bottom of *Figure 1B*; *Figure 1—figure supplement 1B*; *Supplementary file 2B*). Henceforth, we refer to the transition from early-to-late stages in microbial composition as the microbial composition shift or simply the composition shift. In this sampling, we obtained, by chance, a 'no-fly' trap; no adult flies were caught in the trap, and no eggs, larvae, or pupae were found in the food at either stage ('No-fly' in *Figure 1B*; 'No-fly trap' in *Supplementary file 2A, B*; refer to 'Collection of fermented bananas and wild *Drosophila*' in Materials and methods for further details). We analyzed food samples from this trap for comparison. The microbiota in the foods significantly differed from that of other foods, with a lower percentage of yeast in the fungal community and a consistently high abundance of Enterobacterales in the bacterial community.

A previous study showed that the presence of larvae influences the community structures of food microbiota (*Stamps et al., 2012*). Therefore, we sought to determine whether the larvae contributed to the composition shift (*Figure 1C*; *Supplementary file 1A, C*). We prepared microbe-containing suspensions using newly collected early-stage foods and introduced them to fresh bananas, with or without GF embryos. We then incubated the bananas and examined whether the microbial composition shifts occurred in them. Remarkably, the composition shifts occurred similarly, irrespective of the presence of larvae, as indicated by the decreased proportions of Saccharomycodaceae yeast and Enterobacterales bacteria, along with the increased proportions of Pichiaceae yeasts as well as LAB and AAB (compare 'Susp' with 'w/o L' or 'w/L' in *Figure 1D*; see also *Supplementary file 2C, D*). These findings suggest that the composition shift observed in the food microbiota is unlikely to be solely attributed to the presence of larvae but more plausibly influenced by factors such as interspecies interactions among microbes.

Furthermore, we analyzed the microbial composition of adult flies captured in traps, as well as the food and larval samples (*Figure 1E*; *Supplementary file 1A, D*). This analysis revealed a similar composition of the yeast species in adult flies and early-stage foods, and a similar composition of yeast species in larvae and late-stage foods (top of *Figure 1F*; *Supplementary file 2C*). However, notable dissimilarities were observed in the bacterial compositions between the *Drosophila* samples and the early-stage foods, primarily attributed to the conspicuous presence of *Wolbachia* sp. (Anaplasmataceae) and *Gilliamella apicola* (Orbaceae) in the *Drosophila* samples (bottom of *Figure 1F*; *Supplementary file 2E*). Anaplasmataceae and Orbaceae are intracellular and gastrointestinal symbionts of

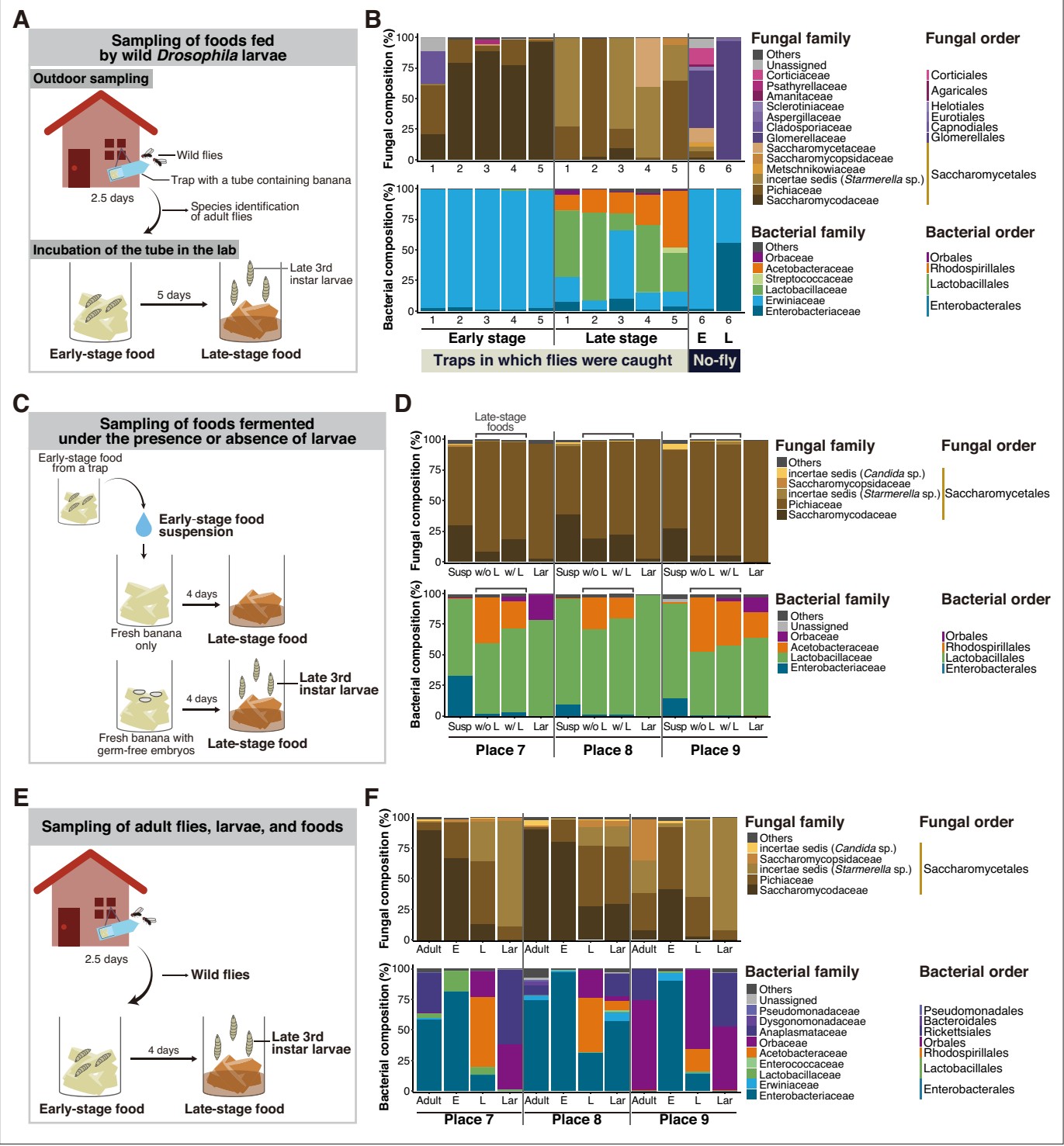

**Figure 1.** The composition of both yeast and bacteria in the food microbiota shifts as fermentation progresses, independently of the presence of larvae. Sampling designs and results for the analyses of microbial compositions in natural food sources of *Drosophila* (**A, B**), effects of larval presence on the microbial composition shift (**C, D**), or microbial compositions in adult flies, larvae, and their foods (**E, F**). (**A, C, E**) Designs of the samplings for microbial analyses. Traps with bananas were set up outdoors near human habitations in order to collect food samples on which wild *Drosophila* spp. lay eggs. Fermented food samples were collected from the traps at two time points: the early stage (just after trap collection) and the late stage (when larvae in the foods developed into late third instars). Collected samples are indicated in bold letters. (**C**) In this sampling, the collected early-stage food was crushed in phosphate-buffered saline (PBS), and the liquid portion (microbial suspension) was added to fresh bananas with or without germ-free embryos (*Cg-Gal4, UAS-mCD8:GFP*). After incubation, late-stage food and surface-sterilized larvae were collected. (**E**) In addition to the

*Figure 1 continued on next page*

*Figure 1 continued*

food samples, adult flies in the traps and larvae from the late-stage foods were collected. This sampling was conducted independently from that shown in (**C**) and (**D**), although the sampling places were in common (see **Supplementary file 1A**). Both adults and larvae were surface sterilized. Note that all of the adults in the traps at Places 7 and 8 were collected for the microbial analysis, while 20 out of 37 were collected at Place 9. (**B, D, F**) The relative abundances of fungi or bacteria in the fermented banana or fly samples. The compositions were analyzed using primer sets amplifying the fungal internal transcribed spacer (ITS) region or bacterial 16S rDNA region. Operational taxonomic units (OTUs) accounting for >1% in any of the samples were grouped by families, as shown. The ratio of those accounting for <1% was summed and is indicated as 'Others'. The genera belonging to uncertain families (incertae sedis) are indicated by their genus name in parentheses. (**B**) The result corresponding to the sampling depicted in (**A**). The numbers on the horizontal axis indicate each numbered sampling location. 'No-fly' indicates the food samples from Sampling Place 6, where no fly or larva was found in the trap or the foods, respectively. (**D, F**) The result corresponding to the sampling depicted in (**C**) and (**E**), respectively. E, early-stage food; L, late-stage food; Susp, microbial suspension; w/o L, late-stage food without larvae; w/ L, late-stage food with larvae; Lar, larvae.

The online version of this article includes the following source data and figure supplement(s) for figure 1:

**Figure supplement 1.** Fungal and bacterial species compositions in natural food sources of *Drosophila*.

**Figure supplement 2.** Copy numbers of microbial rDNA in fermented banana samples.

**Figure supplement 2—source data 1.** Raw data displayed in *Figure 1—figure supplement 2*.

insects, respectively (**Kwong and Moran, 2013**; **Werren et al., 1995**), which plausibly accounts for their relatively lower abundance in the food samples.

In addition to the community structures, we investigated whether there were alterations in the overall abundance of yeasts and bacteria between the early- and the late-stage foods. For this purpose, we performed quantitative PCR to quantify the copy numbers of fungal and bacterial rDNA in each food sample presented in **Figure 1B, D, F** (**Figure 1—figure supplement 2**). The analysis indicated that there were no dramatic increases or decreases in copy numbers in most food samples. Note that measuring rDNA copy numbers in the microbiome does not necessarily reflect actual cell numbers due to variations in the genomic rDNA copy numbers among species (**Lofgren et al., 2019**; **Stoddard et al., 2015**). Nonetheless, these results suggest that the quantities of microbes did not undergo substantial changes between the two stages.

## Prominent acceleration of larval development is observed with early-stage-dominant yeast alone, as well as in combination with late-stage-dominant AAB and other microbes

As described in the previous section, we documented the presence of various yeast and bacterial species in fermented bananas, the populations of which underwent notable compositional shifts over time. Previous studies suggest that different yeast or bacterial species contribute to larval development to varying degrees (**Anagnostou et al., 2010**; **Consuegra et al., 2020b**; **Pais et al., 2018**; **Quan and Eisen, 2018**; **Solomon et al., 2019**). Continuing this line of investigation, we undertook to identify which of the dominating microbial species in the early- or late-stage foods, either individually or as mixtures, promote larval development. To this end, we isolated fungal and bacterial strains from the food samples (refer to **Supplementary file 3** and 'Isolation and species identification of microbes' in Materials and methods for further details). We subsequently inoculated the isolated strains, either individually or in combinations, into a sterile banana-based food (banana agar), and introduced GF larvae. Thereafter, we evaluated the effects of the microbial species on the percentage and the timing of pupariation (see details in 'Quantification of larval development' in Materials and methods). Note that on this food, larval development critically depends on associated microbes, as the larvae failed to pupariate on banana agar without microbes ('GF' in **Figure 2A** and subsequent figures).

To assess the impact of microbes found in the early-stage foods on larval development, we focused on yeast and bacterial strains belonging to microbial species dominated in the food samples shown in **Figure 1B** (see also **Table 1** and **Figure 1—figure supplement 1**). The two characteristic microbial species in the early-stage foods were the yeast *H. uvarum* and the bacteria *Pantoea agglomerans* (**Table 1**; **Figure 1—figure supplement 1**). These two species along with *P. kluyveri*, the second dominant yeast in most of the early-stage foods, were selected for this experiment (**Table 1**; **Figure 1—figure supplement 1A**). We investigated whether each strain or a combination of the strains promotes larval development (**Figure 2A, B**, **Figure 2—figure supplement 1A, B**). When inoculated individually, *H. uvarum* and *P. agglomerans* effectively promoted larval growth ('Y2' and 'E' in **Figure 2A, B**,

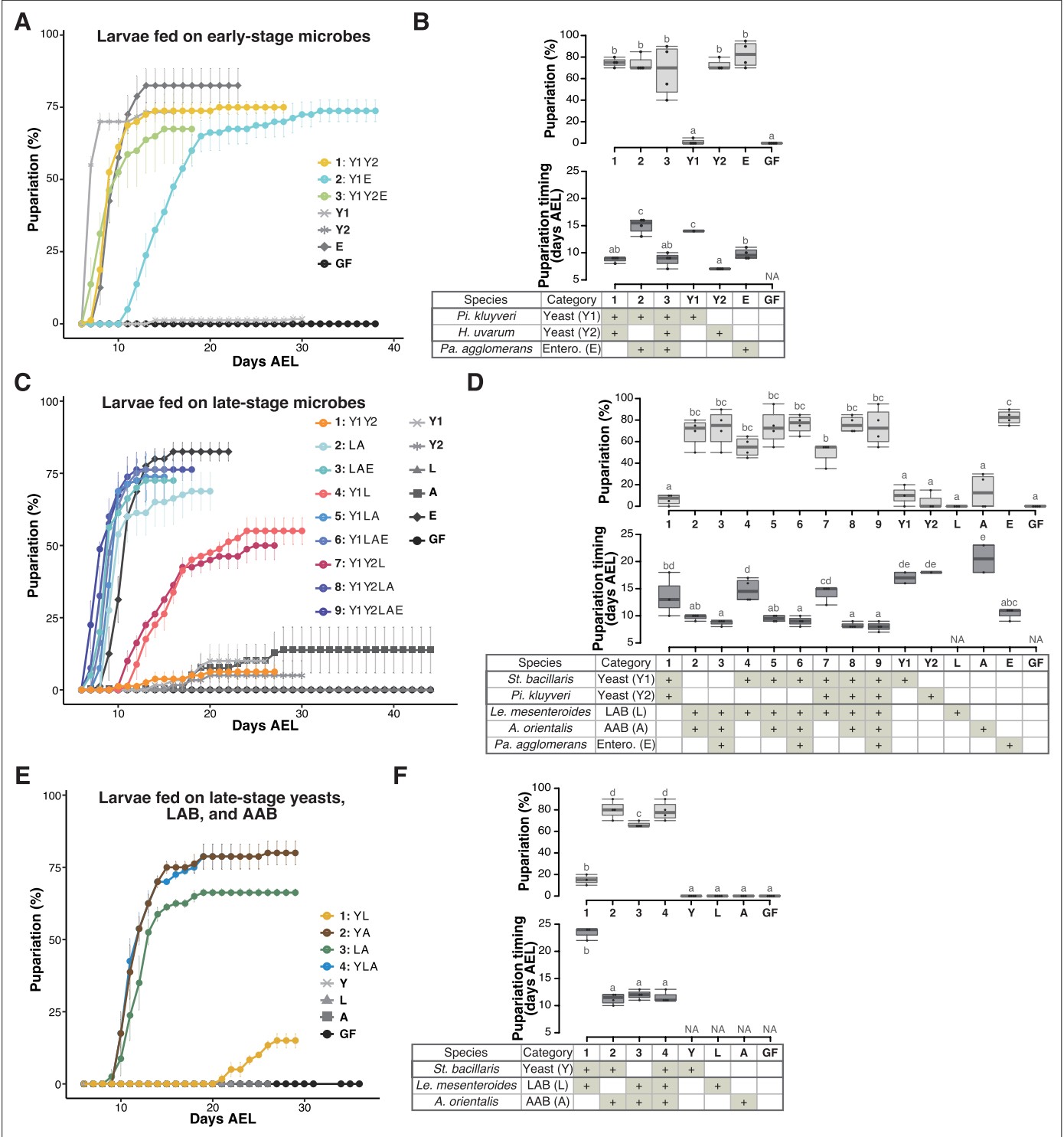

**Figure 2.** Prominent acceleration of larval development occurred with early-stage-dominant yeast alone, as well as in combination with late-stage-dominant acetic acid bacteria (AAB) and other microbes. The percentage and timing of pupariation of larvae feeding on microbes detected in the early-stage food (**A, B**), or five (**C, D**) or three (**E, F**) microbes detected in the late-stage food from Sampling Place 1 in ***Figure 1B***. (**A, C, E**) Each data point represents the average percentage of individuals per tube that pupariated by each day. *n* = 3–4. Colored lines are for mixed conditions, while gray lines are for mono-associated or germ-free conditions. The codes for individual species are provided in the chart at the bottom of (**B**), (**D**), and (**E**), respectively. Error bars represent standard error of the mean (SEM). (**C**) The color coding for the respective foods is as follows: includes yeasts only (orange); includes yeast(s) and lactic acid bacteria (LAB) but no AAB (red); includes LAB and AAB (blue). (**B, D, F**) Percentage of the larvae pupariated (pupariation ratio; upper) and the timing at which 50% of the pupariation ratio was achieved (lower). Boxes represent upper and lower quartiles, while

*Figure 2 continued on next page*

*Figure 2 continued*

the central lines indicate the median. Whiskers extend to the most extreme data points, which are no more than 1.5 times the interquartile range. Unique letters indicate significant differences between groups (Tukey–Kramer test, p < 0.05). *Pi. kluyveri*, *Pichia kluyveri*; *Pa. agglomerans*, *Pantoea agglomerans*; *St. bacillaris*, *Starmerella bacillaris*; *Le. mesenteroides*, *Leuconostoc mesenteroides*; Entero., Enterobacterales; GF, germ-free; NA, not applicable; days AEL, days after egg laying.

The online version of this article includes the following source data and figure supplement(s) for figure 2:

**Source data 1.** Raw data displayed in *Figure 2*.

**Figure supplement 1.** Larvae feeding on mixtures of isolated microbial species detected in the food samples from Sampling Place 2 in *Figure 1B*.

**Figure supplement 1—source data 1.** Raw data displayed in *Figure 2—figure supplement 1*.

**Figure supplement 2.** The percentage of pupariation and eclosion of the larvae feeding on microbes.

**Figure supplement 2—source data 1.** Raw data displayed in *Figure 2—figure supplement 2*.

respectively; 'Y1' and 'E' in *Figure 2—figure supplement 1A,B*, respectively), whereas *P. kluyveri*, another yeast species, had almost no promoting effect ('Y1' in *Figure 2A, B*; 'Y2' in *Figure 2—figure supplement 1A, B*). When the strains were combined, we observed a significantly accelerated larval development in the presence of *H. uvarum* ('1: Y1Y2' and '3: Y1Y2E' in *Figure 2A, B*) than in its absence ('2: Y1E' in *Figure 2A, B*). Hence, in the early-stage foods, *H. uvarum* played a critical role in promoting larval development.

A similar feeding experiment was conducted using strains of the late-stage microbes (*Figure 2C, D* and *Figure 2—figure supplement 1C* and D; *Table 1*): two dominating late-stage yeast species, the most predominant LAB and AAB species, and *P. agglomerans*, which persisted from the early stage, albeit in a smaller proportion (*Table 1*; *Figure 1—figure supplement 1*). When fed individually, none of the yeast or bacterial species efficiently supported larval growth, except for *H. uvarum* and *P. agglomerans* (lines and symbols with different brightness of gray in *Figure 2C* and *Figure 2—figure supplement 1C*). Conversely, when larvae were fed mixtures, their development was almost equally efficient as long as LAB and AAB coexisted (color coded in blues in *Figure 2C* and *Figure 2—figure supplement 1C*). In this experiment, the coexistence effects of yeast and AAB were not tested. Therefore, we repeated the experiment using only one strain from each of the most dominant yeast, LAB, and AAB species (see the tables in *Figure 2F* and *Figure 2—figure supplement 1F*). Larval development, in terms of both the pupariation rate and timing, was promoted when the AAB *A. orientalis* was inoculated together with either a yeast (*S. bacillaris* or *P. kluyveri*) or a LAB (*Leuconostoc mesenteroides* or *Lactiplantibacillus plantarum*) (*Figure 2E, F* and *Figure 2—figure supplement 1E, F*). This result suggests that, in the late-stage foods, the coexistence of AAB and other microbial species, yeasts or LAB, was critical for larval development. Additionally, we investigated the eclosion of pupae under different microbial conditions and confirmed that a majority of the pupae successfully eclosed, albeit with some variation observed across experiments (*Figure 2—figure supplement 2*).

To summarize these results, strong promotion of larval development was observed when the larvae were associated with an early-stage yeast *H. uvarum*, or a combination of *A. orientalis* and other late-stage microbes.

**Table 1.** List of microbial species used in the feeding experiments.
Percentages indicate the relative abundance of each species in the foods shown in *Figure 1B*.

| Yeast or bacterium | Family | Species | Sampling Place 1 (early stage) | Sampling Place 1 (late stage) | Sampling Place 2 (early stage) | Sampling Place 2 (late stage) |
|---|---|---|---|---|---|---|
| Yeast | Saccharomycodaceae | *Hanseniaspora uvarum* | 20.93% | | 79.28% | 2.11% |
| Yeast | Pichiaceae | *Pichia kluyveri* | 40.09% | 26.26% | 19.07% | 97.14% |
| Yeast | incertae sedis | *Starmerella bacillaris* | | 72.97% | | |
| Bacterium | Erwiniaceae | *Pantoea agglomerans* | 45.31% | 12.81% | 90.56% | 6.17% |
| Bacterium | Lactobacillaceae | *Leuconostoc mesenteroides* | | 54.17% | | |
| Bacterium | Lactobacillaceae | *Lactiplantibacillus plantarum* | | | | 60.84% |
| Bacterium | Acetobacteraceae | *Acetobacter orientalis* | | 8.25% | | 15.48% |

## During the late stage, AAB play a crucial role in supporting larval development through interspecies interactions among the microbes

To investigate how larvae respond to the different microbial conditions, we conducted whole-body RNA-seq of gnotobiotic first instar larvae that were subjected to a 15-hr feeding period on banana agar (*Supplementary file 4*). The agar was either kept sterile or inoculated with yeast *H. uvarum*, LAB, or AAB individually, or a combination of LAB and AAB (*Figure 3A–D*). Remarkably distinct gene expression profiles were observed between the 'supportive' conditions, where larvae efficiently pupariated in the previous experiments ('Yeast' and 'LAB + AAB' in *Figure 3A*), and the 'non-supportive' conditions, where larval development was markedly impaired ('LAB' and 'GF' in *Figure 3A*). Notably, a multitude of genes involved in metabolism and cell growth displayed significant upregulation in response to the AAB and LAB mixtures when compared to LAB alone (*Figure 3B, C*, and *Figure 3—figure supplement 1A, B*). Moreover, differentially expressed genes under the supportive and the non-supportive conditions exhibited strikingly similar profiles to those reported in a previous investigation of fed and starved conditions (*Zinke et al., 2002*; *Figure 3D*).

Interestingly, although larvae fed on AAB alone largely failed to pupariate ('A' in *Figure 2C–F*), their gene expression profile after the 15-hr feeding period closely resembled that of the supportive conditions (compare 'AAB' with 'Yeast' and 'LAB + AAB' in *Figure 3A*). This result implies that AAB possessed the ability to induce a growth-promoting response, but the effect likely did not persist until pupariation. We observed that the growth rate of AAB on banana agar was noticeably lower compared to those of other microbes. This led us to speculate that this lower growth rate might result in a shortage of AAB, leading to undernutrition during the later stage of larval development. To test this hypothesis, we investigated whether larval development could recover when AAB was constantly available. Daily supplementation of AAB enabled larvae to pupariate as effectively as the initial co-inoculation of AAB and LAB did (*Figure 3E. F*), demonstrating that AAB can promote larval growth if it is available throughout the course of development.

The aforementioned observations have prompted us to assume that yeast and LAB contribute to the stable coexistence of AAB and larvae. To investigate this hypothesis, we raised larvae on a diet containing AAB either alone or in conjunction with the other species. After a 4-day incubation, we quantified the abundance of AAB in each tube, including those in the food and those inside the larval body. Co-inoculation with other species led to an average 5- to 19-fold increase in the number of AAB colonies compared to the monoculture conditions (*Figure 3G, H*). These findings suggest that AAB plays a crucial role in providing nutrients during the late stages of larval development, while other microbial species support a stable association between AAB and larvae.

As described in the previous section, none of AAB, LAB, or the yeast in the late-stage foods strongly promoted larval growth individually, while mixing AAB with LAB or the yeasts did (*Figure 2C–F* and *Figure 2—figure supplement 1C–F*). All of our results so far strongly suggest that these interspecies interactions among the microbes underlie the promotion of larval growth by late-stage microbiota.

## The isolated yeast species promote varying degrees of larval development, but all support larval development upon heat killing

We found that the early-stage yeast *H. uvarum* facilitated larval development, whereas the dominant yeasts in the late stage, *P. kluyveri* and *S. bacillaris*, did not (*Figure 2A–D* and *Figure 2—figure supplement 1A–D*). To elucidate the underlying mechanisms behind these diverse outcomes, we conducted comprehensive comparative analyses with the respective yeasts. We used six yeast species that originated from the fermented bananas (*Kazachstania humilis*, *Martiniozyma asiatica*, and *Saccharomycopsis crataegensis* in addition to the aforementioned three species) and a laboratory strain of the model species *S. cerevisiae*, which was rarely detected in our food samples (*Figure 4*).

When we provided each species to GF larvae, the larvae fed on *H. uvarum* showed the highest percentage of pupariation, while larvae fed on *K. humilis* or *M. asiatica* also pupariated at relatively high rates, albeit with a slower timing for *M. asiatica* (*Figure 4A, B*). We classified these three species as the 'supportive' yeast species. In contrast, larvae fed on *P. kluyveri* or *S. bacillaris*, which are the dominant species during the late stage, showed notably low percentage of pupariation, leading to their classification as 'non-supportive' (*Figure 4A, B*). Inoculation with *S. crataegensis* or a laboratory strain of *S. cerevisiae* BY4741 resulted in reduced percentages and delayed timing of pupariation

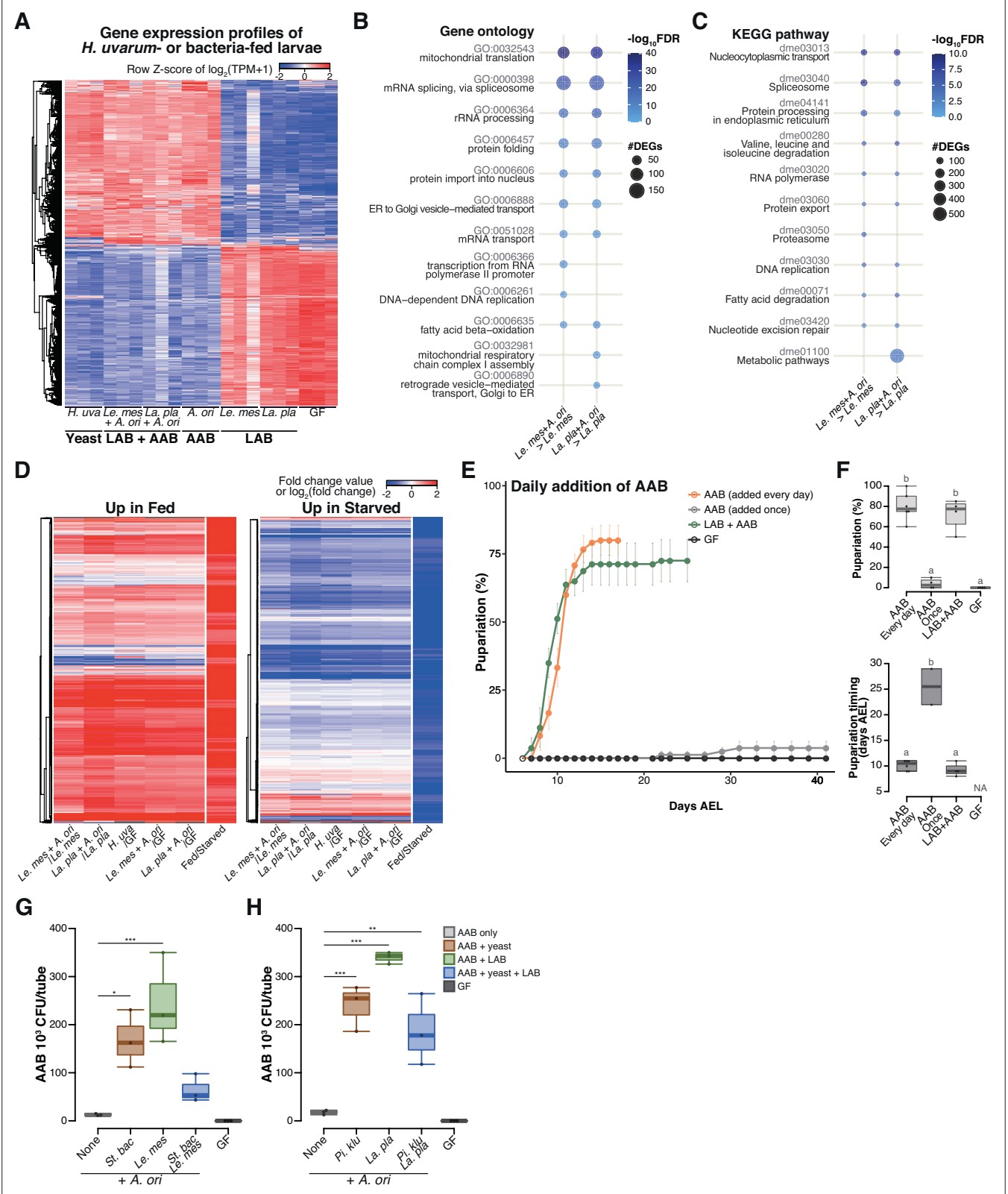

**Figure 3.** During the late stage, acetic acid bacteria (AAB) play a crucial role in supporting larval development through interspecies interactions among the microbes. (**A**) Heat map of gene expression values for first instar larvae fed *H. uvarum* or bacteria. Freshly hatched germ-free larvae were placed on banana agar inoculated with each microbe and collected after 15 hr feeding to examine gene expression of the whole body. The data of the genes that were differentially expressed between 'LAB'-fed conditions and the respective 'LAB + AAB' conditions are shown. Plots showing the result of

*Figure 3 continued on next page*

*Figure 3 continued*

Gene Ontology (GO) term (**B**) or Kyoto Encyclopedia of Genes and Genomes (KEGG) pathway (**C**) enrichment analysis of the RNA-seq data. 10 terms/pathways that showed the smallest false discovery rate (FDR) in each analysis are shown. (**D**) Heat maps showing the similarity between our RNA-seq data and microarray data in a previous study (*Zinke et al., 2002*). Zinke et al. used larvae at 47–49 hr AEL and fed them yeast paste or starved them for 12 hr before comparing the gene expression profiles. The data from *Zinke et al., 2002* are labeled as 'Fed/Starved' in the righthand column of each heat map, which show the fold change value of each gene. Only the genes exhibiting significant differences in their analysis are shown. For our data, $\log_2$(fold change) were calculated using TPM of each gene in the larvae on the supportive conditions versus those on the non-supportive conditions. (**E, F**) The percentage and timing of pupariation of the larvae feeding on AAB on banana agar. Graphs are presented as in *Figure 2*. $n$ = 3–4. (**G, H**) AAB load of foods inoculated with AAB, with or without other microbial species. Boxplots are depicted as in *Figure 2* (Dunnett's test, *p < 0.05, **p < 0.01, ***p < 0.001). *La. pla, Lactiplantibacillus plantarum*; *Le. mes, Leuconostoc mesenteroides*; *A. ori, Acetobacter orientalis*; *H. uva, Hanseniaspora uvarum*; *St. bac, Starmerella bacillaris*; *Pi. klu, Pichia kluyveri*; GF, germ-free; NA, not applicable; days AEL, days after egg laying.

The online version of this article includes the following source data and figure supplement(s) for figure 3:

**Source data 1.** Raw data displayed in *Figure 3*.

**Figure supplement 1.** Enrichment analyses of the genes upregulated in non-supportive conditions and morphology of yeast colonies grown on a nutrient-rich diet.

**Figure supplement 1—source data 1.** Raw data displayed in *Figure 3—figure supplement 1*.

**Figure supplement 1—source data 2.** Raw data of the images displayed in *Figure 3—figure supplement 1E*.

---

compared to the inoculation with the supportive species, thus earning the designation of 'mild'. These results underscore the variable abilities of individual yeast species to promote larval development.

To investigate the response of larvae to different yeast species, we conducted whole-body RNA-seq analysis of larvae (*Supplementary file 5*). The overall findings closely resembled those obtained from various bacterial species (*Figure 4C–F*; *Figure 3—figure supplement 1C, D*; *Figure 3A–D*, *Figure 3—figure supplement 1A, B*). Gene expression profiles exhibited marked differences between larvae fed on the supportive yeasts and those fed on the non-supportive yeasts, with intermediate responses observed in larvae fed on the mild yeasts (*Figure 4C*). Feeding on the supportive yeasts upregulated genes involved in metabolism and cell growth (*Figure 4D, E*), and the expression profiles under the supportive and the non-supportive yeast diets resembled those observed in fed and starved conditions, respectively (*Figure 4F*).

One potential factor determining the ability of larvae to develop on specific yeast species might be the production or secretion of adequate nutrients for larval growth. To assess whether each yeast species produced sufficient essential nutrients internally, we administered heat-killed yeasts to the larvae. Somewhat surprisingly, strong growth enhancement was observed in all seven yeast species upon heat killing, with *P. kluyveri* and *S. bacillaris* promoting larval development nearly as effectively as the supportive yeasts (*Figure 4G, H*). We further explored the effect of killing the yeasts using banana agar supplemented with antifungal agents (butyl *p*-hydroxybenzoate and propionic acid), and observed similar growth promotion by *P. kluyveri* and *S. bacillaris* (data not shown). This finding suggests that all of these yeasts do indeed produce the requisite nutrients for larval development; however, it is likely that the nutrients produced by the live non-supportive yeasts are inaccessible to larvae (further analyzed in the subsequent section).

We also considered the possibility that the non-supportive yeasts somehow inhibited the host growth. To test this possibility, we cultured the yeast species on a nutrient-rich sterile medium and fed them to the larvae (*Figure 4I, J*). This medium contains dry yeast and enables larvae to pupariate even without live yeasts ('GF' in *Figure 4I, J*). Under these conditions, larvae fed with *S. bacillaris* pupariated as efficiently and nearly as rapidly as those without yeasts or with other yeast species, suggesting that *S. bacillaris* did not impede larval growth (*Figure 4I, J*). In contrast, larvae fed with *P. kluyveri* exhibited a significantly reduced percentage of pupariation compared to larvae grown with the other yeast species or without yeasts. This could be related to the extensive growth of *P. kluyveri* on this food (*Figure 3—figure supplement 1E*). Nevertheless, none of these yeast species reproduced the low percentage of pupariation of larvae observed on banana agar. Therefore, the inhibitory effects of *P. kluyveri* and *S. bacillaris* are less likely.

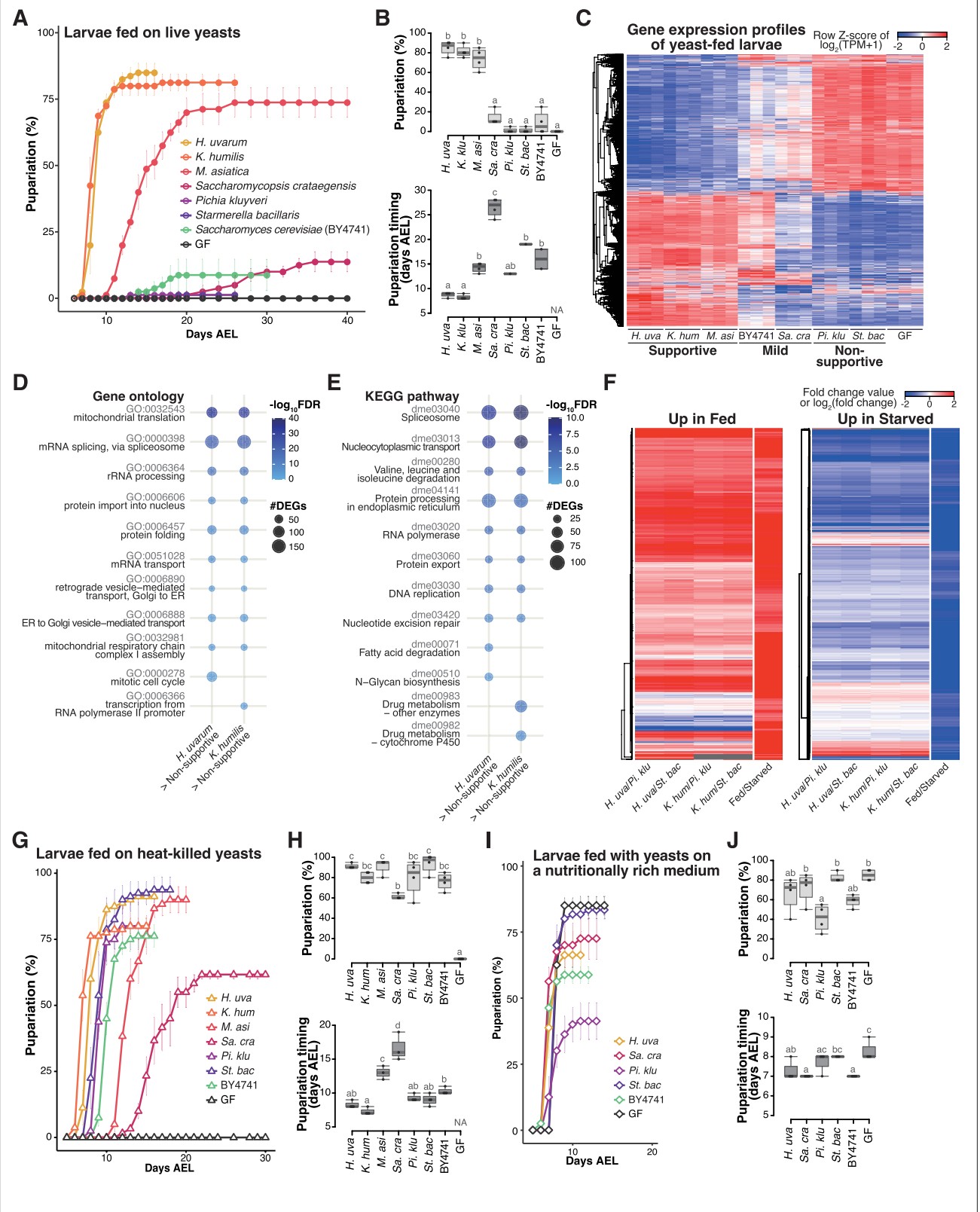

**Figure 4.** Isolated yeast species promote larval development to varying degrees, but all support larval development upon heat killing. (**A, B**) The percentage and timing of pupariation of larvae feeding on live yeasts on banana agar. Graphs are presented as in *Figure 2*. n = 3–4. (**C**) Heat map of gene expression values for first instar larvae fed on each yeast species. Freshly hatched germ-free larvae were placed on banana agar inoculated with each microbe and collected after 15 hr feeding to examine gene expression of the whole body. The data of the genes that were differentially expressed

*Figure 4 continued on next page*

*Figure 4 continued*

between the *H. uvarum*-fed larvae and the *Starmerella bacillaris*-fed larvae are shown. Plots showing the result of GO term (**D**) or KEGG pathway (**E**) enrichment analysis of the RNA-seq data. Genes that showed significantly higher expression on each of the supportive species than on both the non-supportive species were analyzed. 10 terms/pathways that showed the smallest FDR in each analysis are shown. (**F**) Heat maps showing the similarity between our RNA-seq data and microarray data in *Zinke et al., 2002*, shown as described in *Figure 3*. (**G–J**) The percentage and timing of pupariation of the larvae feeding on heat-killed yeasts on banana agar (**G, H**) or live yeasts on a nutritionally rich medium (**I, J**). Graphs are depicted as in *Figure 2*. n = 3–4. *H. uva*, *Hanseniaspora uvarum*; *K. hum*, *Kazachstania humilis*; *M. asi*, *Martiniozyma asiatica*; *Sa. cra*, *Saccharomycopsis crataegensis*; *Pi. klu*, *Pichia kluyveri*; *St. bac*, *Starmerella bacillaris*; BY4741, *Saccharomyces cerevisiae* BY4741 strain; GF, germ-free; NA, not applicable; days AEL, days after egg laying.

The online version of this article includes the following source data for figure 4:

**Source data 1.** Raw data displayed in *Figure 4*.

## Supportive yeasts facilitate larval growth by producing nutrients, including BCAAs, and releasing them from their cells

Given that the non-supportive yeast species supported larval growth upon heat killing, we hypothesized that the key distinction between the supportive and the non-supportive species lies in their ability to release nutrients contained within the cells. To test this hypothesis, we conducted a metabolomic analysis of the two supportive species (*H. uvarum* and *K. humilis*) and the two non-supportive species (*P. kluyveri* and *S. bacillaris*) and analyzed not only the yeast cells but also two additional samples anticipated to contain metabolites released from the cells: yeast-conditioned banana-agar plates and cell suspension supernatants (*Figure 5A–F* and *Figure 5—figure supplement 1*, *Supplementary file 6*).

The yeast-conditioned banana-agar plates had been expected to contain yeast-derived nutrients. On the contrary, the result revealed a depletion of various metabolites originally present in the sterile banana agar (*Figure 5A*). This result prompted us to focus on the metabolites in the chemically defined (holidic) medium for *D. melanogaster* (*Piper et al., 2014*; *Piper et al., 2017*). This medium contains ~40 known nutrients, and supports the larval development to pupariation, albeit at the half rate compared to that on a yeast-containing standard laboratory food (*Piper et al., 2014*; *Piper et al., 2017*). Therefore, the holidic medium could be considered to contain the minimal essential nutrients required for larval growth. Our analysis indicated a substantial reduction of these known nutrients in the yeast-conditioned plates compared to their original quantities (*Figure 5B*).

The quantities of metabolites within the cells varied markedly among the species (*Figure 5C, D*); while the supportive *H. uvarum* contained numerous known metabolites, it was intriguing that *K. humilis*, another supportive species, did not possess as much essential nutrients as other species, including the non-supportive *P. kluyveri* and *S. bacillaris* (*Figure 5D*). This result, in conjunction with the observations from feeding heat-killed yeasts (*Figure 4G, H*), raises the possibility that the interspecies disparities in metabolites within living yeast cells do not directly influence larval growth. On the other hand, our analysis of the cell suspension supernatant revealed distinct variations in the known nutrients in the holidic medium between the supportive and non-supportive species (*Figure 5E, F*). Among these, we focused on BCAAs, leucine and isoleucine (*Figure 5F*). Previous studies have demonstrated that bacteria associated with *Drosophila* provide these essential amino acids to the hosts feeding on artificial diets lacking them (*Henriques et al., 2020*; *Kim et al., 2021*). In our analysis, suspension supernatants of supportive yeasts had concentrations of both leucine and isoleucine that were at least fourfold, on average, higher than those of non-supportive yeasts (*Figure 5F–H*; see also *Supplementary file 6B*).

The above finding prompted us to explore whether leucine and isoleucine are supplied by the associated supportive yeasts, similar to bacterial symbiosis. To investigate this, we supplemented banana agar with these BCAAs and inoculated it with the non-supportive yeasts, subsequently evaluating the restoration of larval development (*Figure 6A, B*). Remarkably, the supplementation of BCAAs elicited a significant improvement in larval development in the presence of the non-supportive yeasts, while it had no effect on larvae fed with the supportive yeasts (*Figure 6A, B*). At 12 days after egg laying, a significant increase was observed in the proportion of individuals that had progressed to the second instar or later stages (*Figure 6B*). These results suggest that the lack of BCAAs in bananas inoculated with the non-supportive yeast species is one of the causes of larval growth deficiency. Additionally, we explored the conservation of the biosynthetic pathways responsible for leucine and isoleucine among

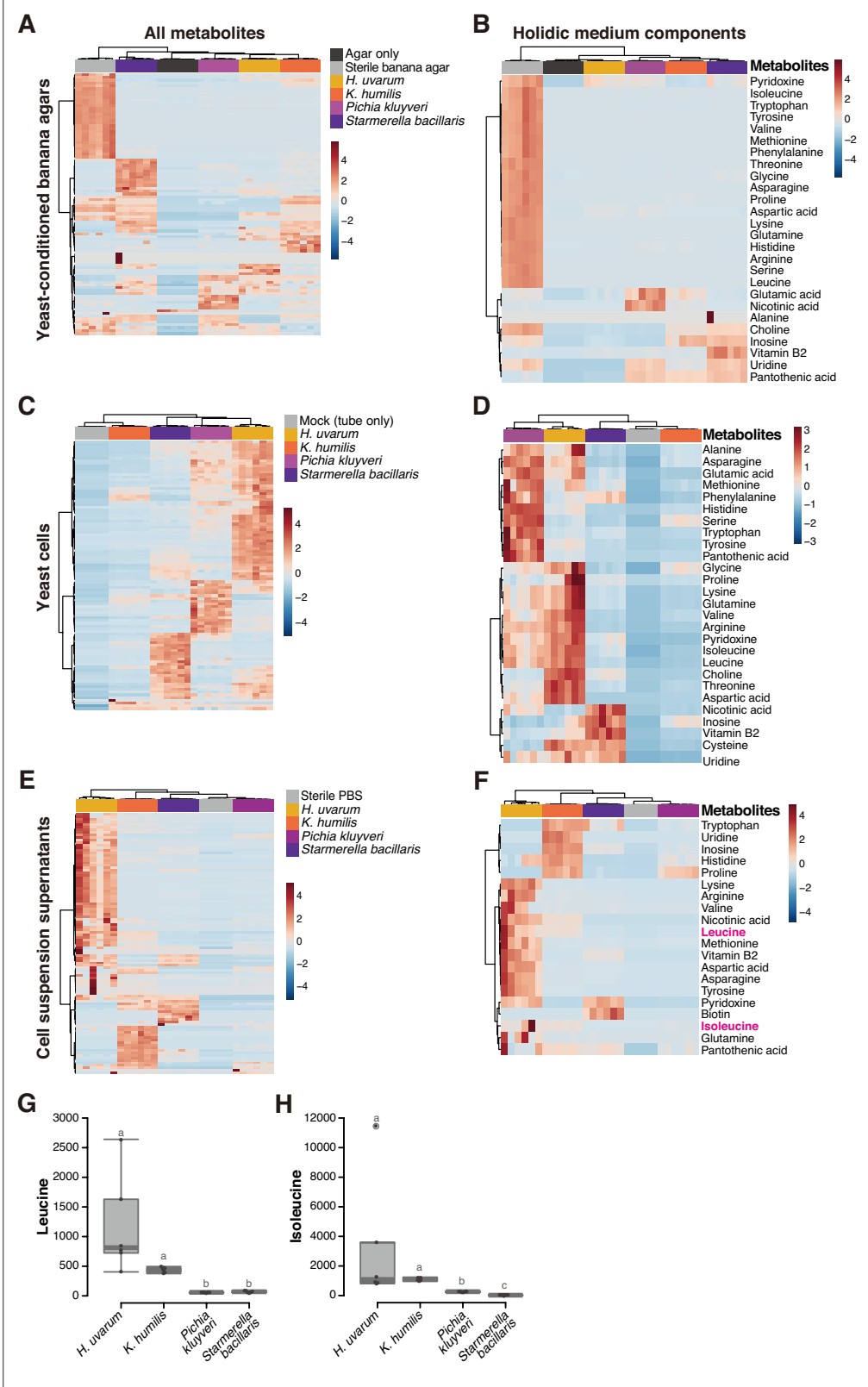

**Figure 5.** Metabolomic analysis of yeast cells, yeast-conditioned banana-agar plates, or cell suspension supernatants. Heat maps displaying all detected metabolites (**A, C, E**) or only the metabolites included in a chemically defined synthetic (holidic) medium for *Drosophila melanogaster* (*Piper et al., 2014*) (**B, D, F**) detected from the metabolomic analysis of banana-agar plates (**A, B**), yeast cells (**C, D**), or yeast-cell suspension

*Figure 5 continued on next page*

*Figure 5 continued*

supernatants (**E, F**). Row *Z*-scores of normalized peak areas are shown. Normalized peak areas of leucine (**G**) and isoleucine (**H**) in yeast-cell suspension supernatants. Boxplots are presented as in *Figure 2*. Unique letters indicate significant differences between groups (Steel–Dwass test, p < 0.05).

The online version of this article includes the following figure supplement(s) for figure 5:

**Figure supplement 1.** Sample preparations for the metabolomic analysis.

the isolated yeast strains. We performed de novo genome assembly and annotation for the six yeast species and examined the presence of orthologs (*Supplementary file 7*). This analysis, along with a subsequent RNA-seq analysis of the yeast species, revealed the conservation of the biosynthetic pathways of these amino acids across all the isolated yeasts (*Figure 6C*; *Supplementary file 8*). The result of this genomic analysis also strengthens the possibility that differences in the ability of yeasts to support larvae do not stem from variances in their ability to biosynthesize BCAAs, but rather in their ability to provide BCAAs in an available form to the larvae.

We noted, however, that the supplementation of BCAAs alone did not completely restore the growth of larvae that fed on the non-supportive yeasts. Despite attempts to enhance growth by increasing the BCAA concentration five- or tenfold, no improvement was observed (data not shown). We also supplemented the banana agar with other metabolites, including nicotinic acid and/or lysine and asparagine, which were detected in higher amounts within the suspension supernatants of the supportive yeasts, individually or in combination with the BCAAs. None of these additions had any effect on larval growth (data not shown). These results suggest that other crucial nutrients are provided by the supportive yeasts.

Regarding the mechanism of nutrient release, one possibility is their secretion from live cells, while another possibility is their leakage from dead cells. To address the latter hypothesis, we collected cells of the four yeast species from banana-agar plates and stained dead cells using Phloxine B (*Figure 6D–G*). Noticeably more dead cells were found in the supportive *K. humilis* compared to other species (*Figure 6E*), implying that nutrients leak from dead *K. humilis* cells into the food, which is subsequently utilized by the larvae.

## Discussion

In this study, we have shown that the microbial composition of larval foods of domestic *Drosophila* species shifts dramatically as fermentation progresses. Individual microbial species play a crucial and indispensable role in promoting host growth, either through the provision of essential nutrients or by establishing a stable association between the host and the nutrient-providing species (*Figure 6H*). The capacity of yeasts to facilitate larval growth seems to rely on their ability to extracellularly release essential nutrients, such as leucine and/or isoleucine, thereby making these nutrients accessible to the host. These microbial functions have effectively empowered the host to grow on a nutritionally inadequate fruit.

*Drosophila* has been known to coexist with yeast and bacteria in nature (*Chandler et al., 2011*; *Chandler et al., 2012*; *Corby-Harris et al., 2007*; *Cox and Gilmore, 2007*; *Quan and Eisen, 2018*; *Shihata and Mrak, 1952*). Our investigation reveals substantial shifts in both fungal and bacterial compositions during fermentation, from the initial predominance of *H. uvarum* to the compositional shifts leading to the dominance of AAB, LAB, and the late-stage yeasts. This shift coincides with alterations in the species serving as nutrient sources. Importantly, most of the microbes we isolated and investigated in this study are generally recognized as associates of *Drosophila*. For instance, yeasts such as *H. uvarum*, *P. kluyveri*, and *S. bacillaris*, or bacteria belonging to the genera *Pantoea*, *Lactiplantibacillus* (formerly *Lactobacillus*), *Leuconostoc*, and *Acetobacter* have also been detected in or isolated from *Drosophila* spp. collected in North America, Europe, or Oceania (*Chandler et al., 2011*; *Chandler et al., 2012*; *Pais et al., 2018*; *Solomon et al., 2019*). These findings suggest the universality of the *Drosophila*–microbe relationships revealed in our study. A previous study reported that *Drosophila* larvae have profound effects on the community structure of yeasts in fermented bananas (*Stamps et al., 2012*). Our experiment, however, suggests that the microbial composition shift observed in our food samples is plausibly influenced by factors such as interspecies interactions

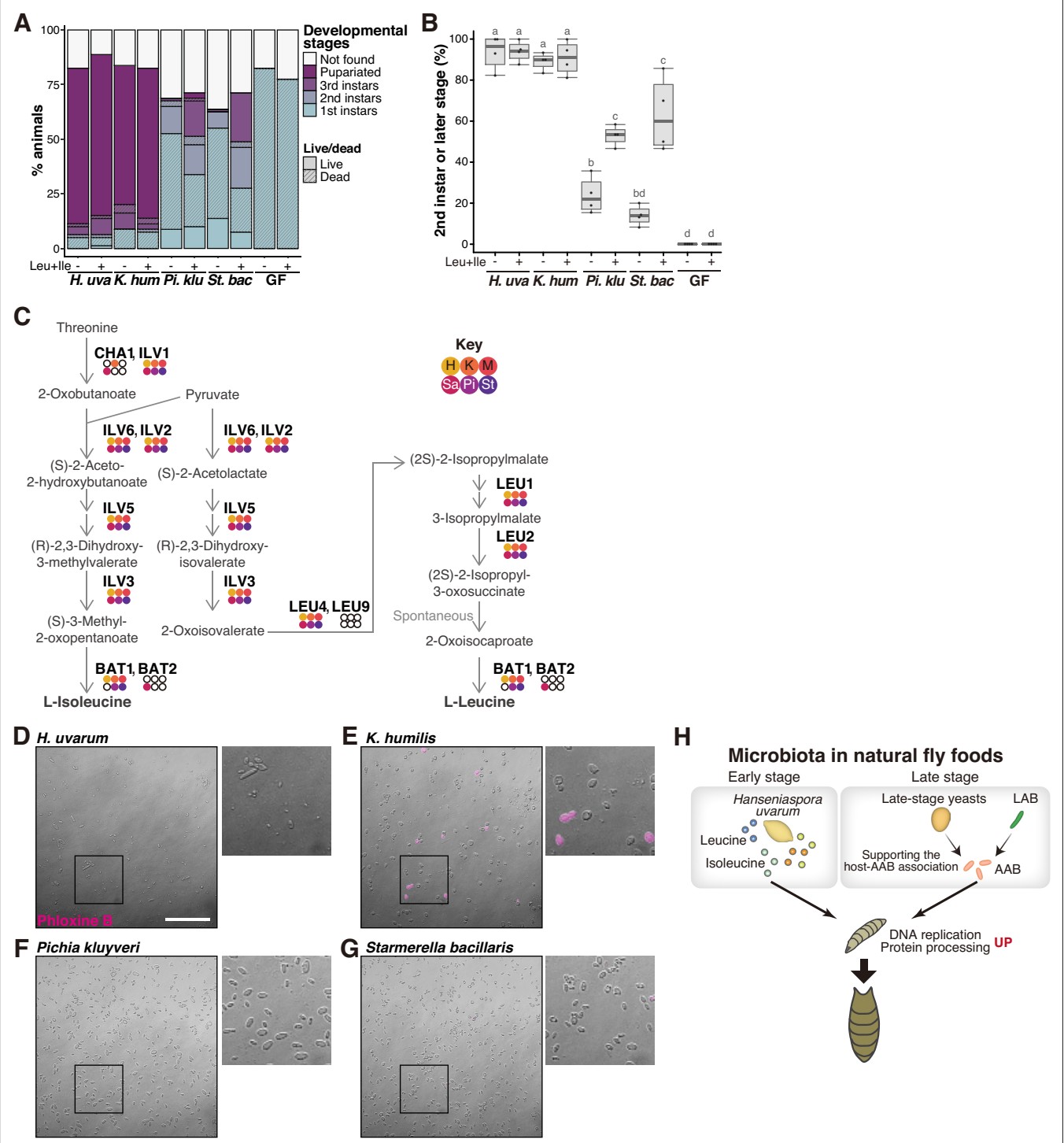

**Figure 6.** Supportive yeasts facilitate larval growth by providing nutrients, including branched-chain amino acids, by releasing them from their cells. (**A, B**) Growth of larvae feeding on yeasts on banana agar supplemented with leucine and isoleucine. (**A**) The mean percentage of the live/dead individuals in each developmental stage. $n = 4$. (**B**) The percentage of larvae that developed into second instar or later stages. The 'Not found' population in (**A**) was omitted from the calculation. Each data point represents data from a single tube. Unique letters indicate significant differences between groups (Tukey–Kramer test, $p < 0.05$). (**C**) The biosynthetic pathways for leucine and isoleucine with *Saccharomyces cerevisiae* gene names are shown. The colored dots indicate enzymes that are conserved in the six isolated species, while the white dots indicate those that are not conserved. Abbreviations of genera are given in the key in the upper right corner. LEU2 is deleted in BY4741. (**D–G**) Representative image of Phloxine B-stained yeasts. The right-side images are expanded images of the boxed areas. The scale bar represents 50 μm. (**H**) Summary of this study. *H. uvarum* is predominant in the early-stage food and provides Leu, Ile, and other nutrients that are required for larval growth. In the late-stage food, AAB directly provides nutrients,

*Figure 6 continued on next page*

*Figure 6 continued*

while lactic acid bacteria (LAB) and yeasts indirectly contribute to larval growth by enabling the stable larva–AAB association. The host larva responds to the nutritional environment by dramatically altering gene expression profiles, which leads to growth and pupariation. *H. uva, Hanseniaspora uvarum*; *K. hum, Kazachstania humilis*; *Pi. klu, Pichia kluyveri*; *St. bac, Starmerella bacillaris*; GF, germ-free.

The online version of this article includes the following source data for figure 6:

**Source data 1.** Raw data displayed in *Figure 6*.

**Source data 2.** Raw data of the images displayed in *Figure 6D–G*.

among microbes. These discrepancies may be due in part to differences in experimental conditions possibly including larval densities, food sample volumes, or culture vessels.

Another implication from our microbial composition analysis is a possible composition shift occurring in the adult intestine. The microbiota of late-third instar larvae resemble those of the late-stage foods they fed on, whereas those of trap-caught adults are similar to those of the corresponding early-stage foods. This result suggests a potential change in microbial composition during pupal stages or after adult eclosion. Unlike developing larvae, which continually consume fermented foods throughout the feeding stages, adults move among a number of food sources, which could lead to the acquisition of new microbes. Additionally, some bacteria have recently been demonstrated to colonize the adult intestine (*Dodge et al., 2023*; *Pais et al., 2018*), possibly contributing to the formation of adult microbiota through the dominance of microbes with a propensity to colonize the intestine.

As for the origin of food microbiota, the initial microbial inoculation to foods is assumed to be mediated by adult flies (*Broderick and Lemaitre, 2012*). Our study revealed the presence of common microbes between our early-stage foods and adult fly samples (*Figure 1F*; *Supplementary file 2C, E*). Furthermore, the foods from the 'no-fly' trap harbored distinct microbiota compared to those from other traps. The distinction was marked by the presence of *Colletotrichum musae*, which is the causal agent of banana anthracnose and might have been present in fresh bananas, and filamentous fungi, which could have originated from airborne spores (*Figure 1B* and *Figure 1—figure supplement 1A*; *Supplementary file 2A*). These results support the previously proposed microbial inoculation to the foods by adult flies. However, alternative sources of microbes should also be considered. For example, microbes may have been infiltrated in unpeeled bananas through peel injuries, as microbial rDNA was detected in bananas that had not undergone microbial inoculation ('Blank' samples in *Figure 1—figure supplement 2C, D*). In addition, they could be introduced by other insects such as rove beetles and sap beetles, which were found in some of the traps, sharing the same habitat with *Drosophila* spp. Further investigation is required to determine the inoculation routes of the microbes in larval foods.

In the early stage of fermentation, both the predominant yeast and bacterial species were capable of supporting larval growth on their own. However, in stark contrast, at the late stage of fermentation, microbial interactions assumed a crucial role. While AAB provide sufficient nutritional resources to support larval growth, their growth rate on bananas is suboptimal. On the other hand, other associated microbes may lack essential nutrients for larvae but contribute to establishing a stable association between AAB and the larvae. Previous studies have reported interactions between AAB and LAB in artificial foods, where AAB act as a source of nutrients, including amino acids, while LAB provide lactic acid as a substrate for amino acid biosynthesis (*Consuegra et al., 2020a*; *Henriques et al., 2020*). Our findings have demonstrated that LAB also play a supportive role in the symbiotic relationship between larvae and AAB on a fruit, a natural food source of *Drosophila* larvae. It is worth noting that interactions may also occur between the AAB and the late-stage yeasts. We observed the presence of various metabolites not only in the suspension supernatant of the supportive yeast species but also in that of the non-supportive late-stage yeast species. Therefore, the growth of AAB on fruits could be enhanced through the utilization of such metabolites by the bacterium.

In laboratory settings, yeasts are recognized and employed as important nutritional sources for *Drosophila*, providing essential nutrients such as amino acids, vitamins, and fatty acids (*Baumberger, 1917*; *Broderick and Lemaitre, 2012*). In our study, we have demonstrated that during larval growth on their natural food sources, the associated yeasts contribute to the provision of BCAAs, leucine, and/or isoleucine. Of these two amino acids, the availability of leucine acts as an effective regulatory signal for the mTORC1 pathway, as established in both mammals and *Drosophila* (*Fox et al., 1998*; *Gu et al., 2022*; *Kim and Guan, 2019*; *Zhang et al., 2021*). The mTORC1 pathway governs

metabolism and cell growth in response to nutritional conditions. Indeed, the expression levels of numerous genes involved in these processes exhibited significant up- or down-regulation in larvae fed on supportive microbes compared to those fed on non-supportive microbes or GF individuals. Nonetheless, supplementing these BCAAs alone did not fully restore larval growth, suggesting that there should be other key nutrients. Our metabolomic analysis did not encompass some known metabolites, including lipids. Among these, sterols, substrates for ecdysone biosynthesis, are indispensable for larval growth. However, sterols are unlikely to be lacking in the fermented bananas. In our experiments, *P. agglomerans* and *A. orientalis* did promote larval growth, but most bacteria, including the two aforementioned species, lack the steroid biosynthesis pathway (*Hoshino and Gaucher, 2021*). Therefore, larvae can grow on bananas without bacterial-derived sterols. Deficiency in other lipids is also unlikely, because larvae can grow on the holidic medium for *D. melanogaster* that contains cholesterol as a sole lipid component, and the developmental delay on the medium is rescued by a 'critical element' which is 'not a lipid' (*Piper et al., 2014*). Therefore, microbial nutrients that promote larval growth, whose functions have not yet been identified in this study, may be components other than lipids.

Despite the well-recognized importance of yeast as a source of nutrients, limited studies have reported the detailed process of how these nutrients are utilized by larvae. Our results suggest that for yeasts to serve as nutritional sources, they should not only produce the nutrients, but also release them outside the cells. The amount of nutrients released extracellularly varied among yeast species. This could be explained by varying abilities to secrete metabolites, although yeasts are unlikely to release the nutrients that are also beneficial to themselves. Another possibility suggested for *K. humilis* was that this species is more prone to death on the culture medium, thus releasing more nutrients. Indeed, yeast autolysates are reported to contain various known nutrients, such as proteins, lipids, and B-group vitamins (*Rakowska et al., 2017*). More thorough investigations should be conducted on another supportive yeast, *H. uvarum*, to assess its mortality on banana-agar plates.

Our results indicate that yeast cells are not necessarily digested in the larval gut, which might contradict previous reports. For instance, vegetative cells of *S. cerevisiae* have been reported to be digested in the gut of adult flies (*Coluccio et al., 2008*). However, the digestibility of yeast may vary depending on the species of yeast, diet composition, or developmental stage of the host. Indeed, a study has discussed the possibility that *S. bacillaris* is not digested in the larval gut (*Solomon et al., 2019*). In our study, by focusing on the non-model yeast species associated with wild flies, we revealed part of the mechanism by which wild *Drosophila* larvae utilize yeast as a nutrient source.

To date, the majority of studies on animal-associated yeasts have focused on a limited number of specific species that are either pathogenic or employed for fermenting foods. In this study, we have examined the species composition of yeasts that make up the symbiotic mycobiota of *Drosophila* and analyzed the roles of each predominant species by investigating their interactions with bacteria as well as with the host. To obtain a comprehensive and detailed understanding of the function of animal microbiota, future studies should further explore the contribution of associated yeasts.

## Materials and methods
### *D. melanogaster* strains and culture

*D. melanogaster* Canton-Special (E-10002) strain was obtained from EHIME-Fly *Drosophila* Stocks of Ehime University. This strain was used for all the experiments unless otherwise noted. *Cg-GAL4* was a gift from *Asha et al., 2003*. The stocks were reared on a laboratory standard diet as previously described (*Watanabe et al., 2017*).

Yeast-based nutrient-rich diet was prepared following the instructions at Bloomington Stock Center (available here). Reagents used are described in *Kanaoka et al., 2023*. The preservatives (propionic acid and 10% *p*-hydroxy-benzoic acid methyl ester) were omitted.

### Generation of GF larvae

GF animals were prepared as previously described (*Watanabe et al., 2019*) with minor alterations. Briefly, embryos were collected on apple agar plates topped with yeast paste, and were incubated at 25°C for 12–15 or 14.5–17.5 hr. They were subsequently dechorionated in 50% bleach, followed by

washes with sterile water, 70% ethanol, and sterile water once more. The embryos were placed on sterile agar plates and incubated at 25°C until newly hatched larvae were obtained.

## Collection of fermented bananas and wild *Drosophila*

Larval foods, fermented bananas, were collected using traps placed outdoors in human residential areas to collect domestic *Drosophila* species such as *D. melanogaster* or *D. simulans*. The sampling places, near laboratory members' apartments or houses in Kyoto and Osaka prefectures, Japan, or outside the laboratory at Kyoto University, are listed in *Supplementary file 1A*. Bananas were selected as bait considering their use in previous *Drosophila* studies (*Anagnostou et al., 2010*; *Consuegra et al., 2020b*; *Stamps et al., 2012*). They are also affordable and available year-round, which would be advantageous for subsequent analyses requiring large amounts of fruit-based culture media. Peels of ripening bananas were treated with 70% ethanol and then removed carefully so as not to touch the fruit inside to avoid contamination. The bananas were subsequently cut into pieces with an autoclaved sterile spatula, and placed in a sterile 100-ml centrifuge tube (Iwaki). Each trap was made of an empty milk carton, and contained one tube of banana bait.

Traps were set up for 2.5 days, which duration was determined from pilot experiments; a shorter collection time resulted in a lower likelihood of obtaining traps visited by adult flies, whereas a longer collection time caused overcrowding of larvae as well as deaths of adults from drowning in the liquid seeping out of the fruits. When the traps were collected, wild flies or their eggs/larvae were found in all traps or foods, except for one (the 'no-fly' trap). In the sampling shown in *Figure 1C–F*, rove beetles (Staphylinidae) and sap beetles (Nitidulidae) were found in several foods. These insects appeared to share a niche with wild *Drosophila* in nature, suggesting that the presence of these insects did not interfere with our goal of obtaining larval food samples. The non-drosophilid insects were removed from the foods immediately after the trap collection. 3–5 ml of the foods (early-stage foods) were sampled, and the remaining foods were incubated at 25°C. Four to five days later, when late third instar larvae were seen in all foods except for the one from the no-fly trap, the foods were collected (late-stage foods). At the first sampling, we added GF embryos of GFP-expressing *D. melanogaster* laboratory strain (*Cg-Gal4, UAS-mCD8:GFP*) to obtain *D. melanogaster* larvae that were reared in the fermented food, but we could not sample them because most animals apparently drowned in the juice that seeped out of the fermented bananas.

To obtain early-stage-food suspensions, newly obtained early-stage foods were crushed in auto-claved sterile phosphate-buffered saline (PBS), and after vigorous mixing, the liquid was collected carefully without taking bananas, embryos, or larvae. 200 µl of this microbial suspension (or sterile PBS for Blank) was added to 100 ml tubes containing fresh ripening banana with or without ~200 GF embryos (*Cg-Gal4, UAS-mCD8:GFP*). The bananas were incubated at 25°C for 4 days. Late-stage foods were collected as described above. Larvae collected from the food were examined under a fluorescent microscope to make sure they were derived from the embryos we added and not from contaminating wild embryos. When sterile PBS was added to bananas instead of microbial suspensions in *Figure 1C*, the food showed no apparent change in appearance or odor during the 4-day incubation, and larvae in the food remained at the first instar stage. The amounts of microbes detected in such foods were less than those obtained for the food inoculated with the microbial suspension by 2–4 orders of magnitude ('Blank' in *Figure 1—figure supplement 2C, D*; see details in the legend).

All the food samples were collected after removing embryos and larvae, and snap-frozen. Larvae were surface sterilized using the same procedure as embryo samples before snap-freezing. Adult flies in the traps were collected following different procedures depending on whether their microbiota were analyzed. When the adult samples were solely utilized for species identification, the whole bodies were stored in 100% or 70% ethanol. When we also required samples for microbial analysis, after washing the whole bodies in 70% ethanol, the head, external genitalia, and wings of each fly were removed and stored in 70% ethanol and subsequently used to identify the species. The rest of the body was snap-frozen and later used for the microbial composition analysis. Adult flies collected in each sampling are listed in *Supplementary file 1B–D*. For the traps where adult flies were caught, the species of the drosophilids in them were identified, confirming the presence of either or both *D. melanogaster* and *D. simulans*.

All tools were sterilized prior to use. Frozen samples were stored at −80°C. Adult or larval samples collected from a single trap or food, respectively, were stored as a pool.

## ITS or 16S rDNA sequencing analysis

Food, larval, or adult samples were freeze-dried for 2–3 days, followed by homogenization with 5 mm stainless beads using Shake Master Neo (Biomedical Science). Microbial DNA was extracted using a QIAamp DNA Stool Mini Kit (QIAGEN). 3 and 0.1 mm zirconium beads were used for the initial lysis step.

Library preparation, sequencing, and data analyses were performed by Macrogen Japan (Tokyo, Japan). For fungal analysis, the ITS region was amplified with primers ITS3 (5'-TCGTCGGCAGCG TCAGATGTGTATAAGAGACAGGCATCGATGAAGAACGCAGC-3') and ITS4 (5'-GTCTCGTGGGCT CGGAGATGTGTATAAGAGACAGTCCTCCGCTTATTGATATGC-3'). For bacterial analysis, the V3–V4 region of 16S rRNA was amplified with primers 341F (5'-TCGTCGGCAGCGTCAGATGTGTATAAGA GACAGCCTACGGGNGGCWGCAG-3') and 805R (5'-GTCTCGTGGGCTCGGAGATGTGTATAAG AGACAGGACTACHVGGGTATCTAATCC-3'). The amplicons were sequenced on an Illumina MiSeq sequencer (300 bp paired-end).

The paired-end reads were assembled using FLASH (*Magoč and Salzberg, 2011*), followed by pre-processing and clustering using CD-HIT-OTU (*Fu et al., 2012*; *Li and Godzik, 2006*) with a clustering cutoff of 99%. Taxonomy was assigned using QIIME (*Caporaso et al., 2010*) using the UNITE database (*Nilsson et al., 2019*) for fungal ITS, and the Ribosomal Database Project (RDP) database (*Cole et al., 2014*) for bacterial 16S rDNA (outputs can be found in *Supplementary file 2F-J*), followed by manual correction and reassignment, in which reads originating from *Drosophila* spp. or banana (including chloroplast) were removed, and fungal and bacterial taxonomy was reassigned by NCBI BLAST search against the Nucleotide collection (nr/nt) database or using the RDP sequence match tool, respectively (*Cole et al., 2014*; *Johnson et al., 2008*). Sequences with the highest query cover and identity in blastn, or the highest S_ab score in RDP sequence match were considered as top hits. OTUs that had no top hits with an assigned species name or with top hits that belonged to more than one genus were marked as 'Unassigned'.

## qPCR of ITS or 16S rDNA

Quantification of fungal ITS or bacterial 16S rDNA was performed by quantitative real-time PCR with THUNDERBIRD SYBR qPCR Mix (TOYOBO). DNA extracted from fermented bananas was diluted 10- to 100-fold before use. After an initial denaturation at 95°C for 1 min, 45 cycles of PCR were carried out: 95°C for 15 s, 55°C for 30 s, and 72°C for 50 s. ITS or 16S rDNA amplicons derived from *H. uvarum* or *P. agglomerans*, respectively, were used for the generation of standard curves. The following primers were used:

For fungal ITS (*White et al., 1990*)

ITS3: 5'-GCATCGATGAAGAACGCAGC-3'
ITS4: 5'-TCCTCCGCTTATTGATATGC-3'

For bacterial 16S (*Hugerth et al., 2014*)

341F: 5'-CCTACGGGNGGCWGCAG-3'
805R: 5'-GACTACHVGGGTATCTAATCC-3'

Banana and *Drosophila* rDNAs were expected to be amplified in this qPCR, as they were in the PCR for the ITS and 16S sequencing analysis. The proportion of microbial rDNA within the total amplification products was assumed to remain consistent between the qPCR and the corresponding sequencing analysis, because the template DNA samples and amplified regions were shared between the analyses. Based on this, the copy number of microbial rDNA was estimated by multiplying the qPCR results with the microbial rDNA ratio observed in the ITS or 16S sequencing analysis of each sample.

## Microbial culture

Microbes were cultured on MRS, YPD, PDA, or banana agar. MRS, YPD, and PDA were prepared using MRS Broth (Merck Millipore), YPD medium (Clontech), and Potato Dextrose Broth (Sigma-Aldrich), respectively, following the manufacturers' protocol. Banana agar was prepared as previously described (*Anagnostou et al., 2010*), with the addition of autoclaving after making the banana-agar mixture. Cavendish bananas (*Musa acuminata*) were used. 10% *p*-hydroxy-benzoic acid in 70%

ethanol (prepared as described in *Kanaoka et al., 2023*), propionic acid (Nacalai Tesque), cycloheximide (Wako), or ampicillin (Nacalai Tesque) were added after autoclaving. Food-derived microbes were cultured at 25°C for 2 days, while *S. cerevisiae* was cultured at 30°C for 2 days before use, unless otherwise noted. Images of single colonies were acquired with a digital camera (Visualix STD2) attached to a stereo microscope (Olympus SZX10).

The *S. cerevisiae* BY4741 strain was obtained from NBRP-Yeast, Japan. To investigate underlying mechanisms of differential growth-promoting ability between AAB and LAB, we attempted to use *Lactiplantibacillus plantarum*[WJL] strain and a genetically engineered BCAA-producing *Lacto*[BCAA] strain (*Kim et al., 2021*), both of which were kind gifts from W.-J. Lee. However, growth of *Lacto*[BCAA] strain on our banana agar was too slow, which made it impossible to obtain a sufficient quantity of cells for the feeding experiments.

## Isolation and species identification of microbes

Microbial strains were isolated on MRS, PDA, or banana-agar plates. The fermented bananas were spread on each of these plates. Single colonies were picked and re-streaked on a plate, and after repeating the process once more, uniform colonies were collected, suspended in MRS liquid medium containing 40% glycerol, and stored at −80°C.

To identify the species of each strain, PCR and sequencing were performed. Microbial colonies were directly suspended in water in PCR tubes and heated (96°C for 5 min) before the addition of other reagents. In the cases when DNA failed to amplify, DNA was extracted using InstaGene Matrix (Bio-Rad) according to the manufacturer's protocol. NL1/NL4 primers were used for fungi, while 27F/1492R primers were used for bacteria (*Chandler et al., 2011*; *Chandler et al., 2012*), and KOD FX (TOYOBO) was used for the amplification. The amplicon was purified prior to sequencing using a Wizard SV Gel and PCR Clean-Up System (Promega).

Sequencing was performed using an Applied Biosystems 3130xl Genetic Analyzer, and primers were NL1 for fungi and 518F (5′-CCAGCAGCCGCGGTAATACG-3′) for bacteria (https://www.macrogen-japan.co.jp/cap_seq_0104.php). See *Supplementary file 3A* for the sequencing results.

Species identification of isolated yeast strains was performed by NCBI BLAST searches against the Nucleotide collection (nr/nt) database (*Johnson et al., 2008*). Sequences with Percent Identity = 100%, *E*-value = 0 were considered as top hits, and the species name that appeared in more than two top hits was used to refer to the strain. See also *Supplementary file 3B* for the number of the top hits for each strain.

For bacteria, representative sequences of OTUs from the microbial composition analysis were compared with the sequences of isolated strains, and the strain with the closest 16S rDNA sequence was used as the strain corresponding to that OTU. See also *Supplementary file 3C*.

## Quantification of larval development

The microbes were individually grown on banana agar twice before being suspended in PBS (OD600 = 60). To mix multiple species, equal amounts of the suspensions prepared as described were mixed together. 10 μl of each suspension was added to ~1.1 ml of autoclaved banana agar in 1.5-ml tubes. Tubes were incubated at 25°C for 2 days before larvae were added. The following procedure was performed as described in *Watanabe et al., 2019*, with a few modifications. Briefly, 20 GF larvae were added to each tube, and the tubes were kept in a moisturized incubator (80–90% humidity) at 25°C. Pupariated individuals were removed from the tubes prior to eclosion, and either discarded or transferred to vials for further incubation until eclosion occurred. The number of pupae in each tube was counted until all larvae either pupariated or died. In experiments that require daily addition of AAB, the bacterial cells (a lump of ~3 μl by volume per tube) were added daily. As more individuals pupariated, less bacteria were consumed each day. Therefore, to avoid the excessive accumulation of the bacteria, no AAB was added in a tube where the previously added bacterial lump was visible to the naked eye.

The effect on larval development was quantified as the final percentage of the pupariated individuals and timing of the pupariation. The developmental timing was quantified from the date on which the percentage of pupariation exceeded 50% of the final percentage of pupariation. Molds were occasionally seen on foods during incubation, in which case the tubes were removed from the

experiment. On a rare occasion when the number of samples for a single condition fell below 3, all the tubes for the condition were excluded from the results.

When feeding heat-killed yeasts to larvae, yeasts were added to the banana-agar tubes and subsequently heated as following procedures. The yeasts were revived from frozen stocks on banana-agar plates, incubated at 25°C, and then streaked on fresh agar plates. After 2-day incubation, yeast cells were scraped from the plates and suspended in PBS at the concentration of 400 mg of yeast cells in 500 μl of PBS. 125 μl of the suspensions were added to banana-agar tubes prepared as described, and after centrifugation at 3000 × $g$ for 5 min, the supernatants were removed. The amount of cells in each tube is ~50× compared to that when feeding live yeasts, which compensates for the reduced amount due to their inability to proliferate. The tubes were subsequently heated at 80°C for 30 min before adding GF larvae.

When feeding yeasts on banana agar supplemented with antifungal agents, the yeasts were individually grown on normal banana agar twice before being suspended in PBS at the concentration of 400 mg of yeast cells in 500 μl of PBS. 125 μl of the suspensions was introduced onto the anti-fungal agents (10 ml/l 10% butyl $p$-hydroxy-benzoate in 70% ethanol and 6 ml/l propionic acid, following the concentration described in *Kanaoka et al., 2023*)-containing banana agar in 1.5 ml tubes. After centrifugation, the supernatants were removed. The amount of cells in each tube is ~50× compared to that when feeding live yeasts.

## Sequencing and annotation of yeast genomes

To extract genomic DNA from yeast cells for genome sequencing, yeasts were cultured on MRS plates. Collected cells were treated with Zymolyase-20T (Nacalai Tesque) to remove cell walls, and DNA was extracted using QIAGEN Genomic-tip 20G kit (QIAGEN).

The genomic DNA prepared from six strains was sequenced by both Illumina and Oxford Nanopore Technologies (ONT) technologies. For Illumina sequencing, libraries were prepared with QIAseq FX DNA Library Kit (QIAGEN) and sequenced on the Illumina MiSeq platform (Illumina, CA, USA) to generate 2 × 301 bp paired-end sequence reads. MiSeq reads were trimmed using Platanus_trim (http://platanus.bio.titech.ac.jp/pltanus_trim). The thresholds for quality and read length were set at 15 and 25, respectively. After trimming, a total of 1.863–2.396 Gb sequence was obtained for each genome (average; 2.113 Gb). Sequence coverages estimated by GenomeScope (*Vurture et al., 2017*) were over ×100 for all genomes (×103.42–×210.48). For ONT sequencing, libraries were prepared with Rapid Barcoding Kit (ONT) and sequenced using the R9.4.1 flow cell with the MinION platform, followed by base-calling using Albacore ver. 2.3.3 (ONT). Adapter sequences of MinION reads were trimmed with NanoFilt (*De Coster et al., 2018*) or Porechop (https://github.com/rrwick/Porechop; *Wick, 2018*), followed by filtering with a quality threshold of 10 and a length threshold of 2000. After filtering, 84,598–197,215 reads (765–4402 Mb in total sequence length) were obtained for each genome (average; 121,913 reads and 1035 Mb). Estimated sequence coverages by MinION reads were ×63.1–×107.3 (average; ×87.9).

For RNA-seq analysis, equal amounts of yeast cells cultured under six conditions (two temperatures: 25 and 30°C, and three culture plates: MRS, YPD, or banana-agar plates) were mixed and used for RNA extraction. RNA was extracted as previously described (*Iida and Kobayashi, 2019*) and purified using the RNeasy mini kit (QIAGEN). Libraries for RNA-seq were prepared with NEBNext Ultra II Directional RNA Library Prep Kit for Illumina (NEB) and sequenced on the NextSeq 500 platform (Illumina) to generate 300-base paired end sequences which were used for annotation described below. A total of 23.48–26.75 M reads were obtained for each yeast (average; 25.63 M reads).

Genome assembly was performed by MaSuRCA v3.2.6 (*Zimin et al., 2013*) using MiSeq and MinION reads. In the heterozygosity rate estimation by GenomeScope based on MiSeq reads, the *K. humilis* and *P. kluyveri* genomes showed higher heterozygosity rates (*K. humilis*; 2.98%, *P. kluyveri*; 2.08%) while those of other genomes were 0.02–1.00%. In the assessment of autoannotated genome assemblies using BUSCO (*Simão et al., 2015*), completeness was over 86% in the four genomes other than the *H. uvarum* (64.58%) and *S. bacillaris* (75.21%) genomes. However, among the four genomes, those of *P. kluyveri* and *K. humilis* showed a high proportion of 'Duplicated complete' genes (*K. humilis*; 27.74%, *P. kluyveri*; 5.17%, respectively), which was apparently due to the high heterozygosity rate of these genomes. Therefore, these two genomes were assembled using Platanus-allee v2.2.2 (*Kajitani et al., 2019*), and by comparing it with the MaSuRCA assemblies using GenomeMatcher (*Ohtsubo*

*et al., 2008*), redundant sequences in the MaSuRCA assemblies were identified and removed from the assemblies. As the proportions of 'Duplicated complete' genes estimated by BUSCO were 5.51% for the *K. humilis* genome and 0.74% for the *P. kluyveri* genome after removing redundant sequences, these redundant sequence-removed genome assemblies were used as the final genome assemblies of *P. kluyveri* and *K. humilis*.

Genome annotation was performed using FunGAP (*Min et al., 2017*) with the above-mentioned RNA-seq data obtained from each yeast. Over 91% of RNA-seq reads obtained was mapped to the final assemblies of each genome using Hisat2 (*Kim et al., 2015*). In the assessment of annotated genomes using BUSCO, completeness was >86% for the four genomes other than those of *H. uvarum* (65.95%) and *S. bacillaris* (76.92%). It appears that the low completeness of these genomes is attributable to genome reduction. The final statuses of genome assembly and annotation of each genome are summarized in *Supplementary file 7*.

Mitochondrial sequences in each genome assembly were identified by tblastx homology search with fungal mitochondrial sequences obtained from the RefSeq database (*O'Leary et al., 2016*). When multimerized mitochondrial sequences were generated in genome assemblies, they were trimmed to the smallest unit.

To determine gene orthology between *S. cerevisiae* and each of the isolated strains, blastp was performed. *S. cerevisiae* protein sequence data (GCF_000146045.2_R64_protein.faa) were downloaded from NCBI genome database (https://www.ncbi.nlm.nih.gov/genome). Reciprocal best hits with an *e*-value $<10^{-10}$ were defined as orthologs.

## Quantification of AAB

Microbes were inoculated onto 200 ml of banana agar in a tube included in a Biomasher II homogenizer kit (NIP). After adding five GF larvae, the tubes were incubated at 25°C for 4 days. 150 µl of PBS was added to the tube, and food was crushed along with the larvae in it. More PBS was subsequently added to bring the total volume of the suspension to 700 µl. 1/5 serial dilutions were made, 50 µl of which were spread onto MRS plates supplemented with antibiotics (10 µg/ml cycloheximide and 10 µg/ml ampicillin to inhibit the growth of yeasts and LAB, respectively). The plates were incubated at 25°C, and 3 days later, colonies were counted to calculate CFU.

## RNA sequencing for gene expression analyses

For larval RNA-seq analysis, microbes were inoculated on banana agar as previously described and freshly hatched GF larvae were added. After 15 hr feeding, 20 larvae were collected and snap-frozen. RNA extraction, sequencing, and data analysis were performed as previously described (*Kanaoka et al., 2023*).

Yeast RNA was extracted as described above, and sequenced as previously described in *Kanaoka et al., 2023*. Data obtained from the FunGAP analysis were used for mapping of reads obtained for the isolated strains, and the *S. cerevisiae* S288C genome R64-1-1, retrieved from Illumina iGenomes for *S. cerevisiae* BY4741.

For the heat maps in *Figures 3A and 4C*, heatmap.2 in the gplots (*Warnes et al., 2022*) package of R (*R Development Core Team, 2020*) was used. Database for annotation, visualization and integrated discovery (DAVID) Functional Annotation Chart (*Huang et al., 2009*; *Sherman et al., 2022*) was used to find significantly enriched GO terms or KEGG pathways in each analysis. For comparisons with the microarray analysis result of *Zinke et al., 2002*, their gene expression data (fold change value) of the individuals at 12 hr after feeding or starvation was used. For our data, $\log_2$(fold change) of gene expression was calculated for each supportive and non-supportive condition pair. Genes that were represented by more than one probes were omitted. The FlyBase IDs in the data of *Zinke et al., 2002* were converted to current IDs using FlyBase ID Validator (https://flybase.org/convert/id). Only the genes that had one-to-one correspondence with current FlyBase IDs were included in the heat maps. The genes that showed significantly different expression between fed ('normal' in the cited paper) and starved conditions, that is, those with fold-change values of ≥1 or ≤−1, are shown.

## Preparing samples for LC–MS analysis

The yeasts were individually grown on banana agar three times. 100 mg of cells were scraped from the surface of the plates and snap-frozen. To prepare yeast-conditioned plates, yeasts were grown as

described and thoroughly removed by scraping. 22 mm$^2$ chunks of agar from the plates (~100–160 mg) were collected and snap-frozen. 15 g/l agar plates or sterile banana-agar plates were prepared as controls and collected using the same procedure. To prepare cell suspension supernatants,~250 mg of yeast cells were collected as described, and sterile PBS was added at a ratio of 5 µl PBS per 1 mg of yeast cells. After suspending the cells, the suspensions were centrifuged at 2260 × *g* for 5 min. The supernatants were further filtered using a 0.45-µm Millex-HA filter (Merck) to completely remove any remaining yeast cells. 700 µl of each supernatant was collected and snap-frozen.

Each sample was suspended or diluted to 500µl of methanol containing internal standard; 30 µM 2-Morpholinoethanesulfonic acid and 30 µM L-Methionine. After mixing with 250 µl of water and 400 µl of chloroform, the samples were centrifuged and the upper layer was collected and filtered using an UltrafreeMC-PLHCC for Metabolome Analysis column (Human Metabolome Technologies, #UFC3LCCNB-HMT). The samples were dried completely using nitrogen gas and resuspended in water before injection into the LC–MS system.

## LC–MS/MS measurement

Cationic metabolites including amino acids and nucleosides are quantified using a triple-quadrupole mass spectrometer equipped with an electrospray ionization (ESI) ion source (LCMS-8060, Shimadzu Corporation) in the positive and negative-ESI and multiple reaction monitoring (MRM) modes. The samples were resolved on the Discovery HS F5-3 column (2.1 mm ID × 150 mm, 3 µm particle, Sigma-Aldrich), using a step gradient with mobile phase A (0.1% formate/water) and mobile phase B (0.1% formate/acetonitrile) at ratios of 100:0 (0–5 min), 75:25 (5–11 min), 65:35 (11–15 min), 5:95 (15–20 min), and 100:0 (20–25 min), at a flow rate of 0.25 ml min$^{-1}$ and a column temperature of 40°C. Chromatogram peaks obtained with compound-specific MRM channels were integrated and manually confirmed. If more than one confirmatory MRM channel was available for the target compound, it was set to confirm the identification of the peak compound. Structural isomers were separated either by retention time on the column or by compound-specific MRM signals. Detailed MRM conditions are identical to the previously published study (*Oka et al., 2017*). To further confirm the signal specificity, sample-derived chromatographic peaks were compared with the corresponding standards to ensure that retention times were consistent. Peak quantification values obtained for each compound were corrected for recovery due to L-Met in the IS. Data are presented as peak area values for each metabolite normalized by the internal standard. Those for the yeast-conditioned plates or cells were further normalized with sample wet weight. Heat maps were generated by MetaboAnalystR 3.3.0 (*Pang et al., 2020*), using data for all of the detected metabolites or a subset of metabolites that are included in the holidic medium of *D. melanogaster* (*Piper et al., 2014*).

## Developmental progression with BCAA supplementation

Yeasts were cultured as described on autoclaved banana agar supplemented with 2.03 g/l of leucine (Nacalai Tesque) and 1.12 g/l of isoleucine (Peptide Institute), both at the concentrations included in the holidic medium with exome-matched FLYAA (*Piper et al., 2017*). Twenty larvae were added to each tube, and after 11 days of feeding, live or dead individuals in each developmental stage were counted. Larval developmental stages were determined based on tracheal morphology, as previously described (*Niwa et al., 2010*). The percentage of the larvae that could not be found are indicated as 'Not found'.

Technically, it was difficult to match the amounts of leucine and isoleucine to be added to the foods to the difference in amounts released by the supportive and non-supportive species. This was because under our experimental condition, quantification of the microbe-derived BCAAs accumulating in the culture medium, which was performed in previous studies (*Consuegra et al., 2020a*; *Henriques et al., 2020*), was impractical due to constant absorption of the nutrients by the yeasts themselves. Given this constraint, we opted for the amino acid concentrations in the holidic medium, which concentrations are sufficient to support larval growth under axenic conditions and not detrimental to larval growth (*Piper et al., 2014*).

## Imaging of dead yeast cells

Microbes were grown on banana-agar plates for 3 days, then collected and suspended in PBS at a concentration of ~1 × 10$^8$ cells/ml. The cells were subsequently incubated in 5 µg/ml Phloxine B

(Wako) for 10 min at room temperature. The cells were subsequently washed twice with PBS and observed using a Nikon C1 laser scanning confocal microscope coupled to a Nikon Eclipse E-800 microscope. Note that the brightness of the images was adjusted with a gradation to achieve mostly uniform brightness, because the bottom side of the original images was darker due to microscope malfunction.

## Statistical analysis

R (*R Development Core Team, 2020*) was used for all statistical analyses. p-values less than 0.05 were considered statistically significant. For statistical analyses of RNA-seq data, see *Supplementary file 4*, *Supplementary file 5*, and *Supplementary file 8*. See *Supplementary file 9*, Materials and methods, and figure legends for analysis of other experiments.

## Acknowledgements

We thank T Kondo and Y Sando for performing RNA sequencing; M Umeda and T Suito for technical advice on microbial isolation and identification; K Furuya for technical advice on yeast genomic DNA extraction and Phloxine B staining; T Iida and T Kobayashi for technical advice on yeast RNA extraction; J Hejna for polishing the manuscript; H Imai, K Oki, and M Futamata for technical and secretarial assistance; T Kuraishi, A Hori, Y Degawa, Y Yoshihashi, R Niwa, T Ito, Y Sakai, H Yurimoto, and K Shiraishi for technical advice and discussion; W-J Lee for providing bacterial strains; and members of the Uemura laboratory for technical advice, discussion, and kind cooperation in obtaining fermented food samples. This work was supported by the Japan Society for the Promotion of Science (JSPS; 21K06186, 17K15039, and 16H06279 [PAGS] to YH, 17KT0018 to TU, and 20J23332 to AM); AMED-CREST (JP18gm1110001 to TU); and JST FOREST Program (JPMJFR2051 to YH).

## Additional information

### Funding

| Funder | Grant reference number | Author |
|---|---|---|
| Japan Society for the Promotion of Science | 21K06186 | Yukako Hattori |
| Japan Society for the Promotion of Science | 17K15039 | Yukako Hattori |
| Japan Society for the Promotion of Science | 16H06279 | Yukako Hattori |
| Japan Society for the Promotion of Science | 17KT0018 | Tadashi Uemura |
| Japan Society for the Promotion of Science | 20J23332 | Ayumi Mure |
| Japan Agency for Medical Research and Development | JP18gm1110001 | Tadashi Uemura |
| Fusion Oriented REsearch for disruptive Science and Technology | JPMJFR2051 | Yukako Hattori |

The funders had no role in study design, data collection, and interpretation, or the decision to submit the work for publication.

### Author contributions

Ayumi Mure, Resources, Data curation, Formal analysis, Funding acquisition, Investigation, Visualization, Methodology, Writing - original draft; Yuki Sugiura, Resources, Formal analysis, Investigation, Visualization, Methodology, Writing - original draft; Rae Maeda, Formal analysis, Investigation, Visualization, Methodology, Writing - original draft; Kohei Honda, Formal analysis; Nozomu Sakurai,

Investigation, Methodology; Yuuki Takahashi, Aina Gotoh, Methodology; Masayoshi Watada, Formal analysis, Investigation, Methodology, Writing - review and editing; Toshihiko Katoh, Investigation, Methodology, Writing - review and editing; Yasuhiro Gotoh, Itsuki Taniguchi, Keiji Nakamura, Data curation, Formal analysis, Investigation, Methodology; Tetsuya Hayashi, Data curation, Formal analysis, Investigation, Methodology, Writing - original draft; Takane Katayama, Resources, Investigation, Methodology, Writing - review and editing; Tadashi Uemura, Conceptualization, Supervision, Funding acquisition, Methodology, Writing - original draft, Writing - review and editing; Yukako Hattori, Conceptualization, Formal analysis, Supervision, Funding acquisition, Methodology, Writing - original draft, Project administration, Writing - review and editing

**Author ORCIDs**
Ayumi Mure ⓘ http://orcid.org/0009-0006-8922-6816
Yasuhiro Gotoh ⓘ http://orcid.org/0000-0001-9716-2820
Tetsuya Hayashi ⓘ http://orcid.org/0000-0001-6366-7177
Takane Katayama ⓘ http://orcid.org/0000-0003-4009-7874
Tadashi Uemura ⓘ http://orcid.org/0000-0001-7204-3606
Yukako Hattori ⓘ http://orcid.org/0000-0001-5977-8501

Reviewer #1 (Public Review): https://doi.org/10.7554/eLife.90148.3.sa1
Reviewer #2 (Public Review): https://doi.org/10.7554/eLife.90148.3.sa2
Reviewer #3 (Public Review): https://doi.org/10.7554/eLife.90148.3.sa3
Author Response https://doi.org/10.7554/eLife.90148.3.sa4

## Additional files

**Supplementary files**
• Supplementary file 1. Details of the samplings of fermented bananas and wild *Drosophila*.
• Supplementary file 2. Microbial composition analysis results.
• Supplementary file 3. Microbial isolation and species identification.
• Supplementary file 4. RNA-seq data of whole bodies of first instar larvae fed on *H. uvarum* or bacteria.
• Supplementary file 5. RNA-seq data of whole bodies of first instar larvae fed on yeasts.
• Supplementary file 6. Results of metabolomic analysis of yeast samples.
• Supplementary file 7. Final genome assembly and annotation statuses of six yeast strains.
• Supplementary file 8. RNA-seq data of isolated yeasts.
• Supplementary file 9. Statistical details of experiments.
• MDAR checklist

**Data availability**
All sequence data obtained in this study have been deposited in the DDBJ Sequence Read Archive. The accession numbers for the sequence data are DRR212820–DRR212838 (BioProject accession number: PRJDB9099) and DRR471830–DRR471964 (BioProject accession number: PRJDB15711). Yeast genome annotation data have been deposited in DDBJ/EMBL/GenBank under BioProject accession number PRJDB9099. The accession numbers are BTGD01000001–BTGD01000029 (Kazachstania humilis), BTGA01000001–BTGA01000014 (Hanseniaspora uvarum), BTFZ01000001–BTFZ01000021 (Saccharomycopsis crataegensis), BTGB01000001–BTGB01000010 (Pichia kluyveri), BTFY01000001–BTFY01000014 (Martiniozyma asiatica), and BTGC01000001–BTGC01000009 (Starmerella bacillaris). All data generated or analyzed during this study are included in the manuscript and supporting files; Source Data files have been provided for Figures 2, 3, 4, and 6, and Figure 1—figure supplement 2, Figure 2—figure supplement 1, Figure 2—figure supplement 2, and Figure 3—figure supplement 1. Figure 6—Source Data 2 and Figure 3—figure supplement 1— Source Data 2 contain the original data of the images in each figure. Other Source Data files contain the numerical data used to generate the figures. Numerical data for other figures are provided in Supplementary Files.

The following datasets were generated:

| Author(s) | Year | Dataset title | Dataset URL | Database and Identifier |
|---|---|---|---|---|
| Hattori Y, Mure A, Uemura T | 2023 | Genome and transcriptome sequencing of 6 *Drosophila*-associated yeast species | https://ddbj.nig.ac.jp/resource/bioproject/PRJDB9099 | DDBJ, PRJDB9099 |
| Mure A, Hattori Y | 2023 | ITS or 16S amplicon sequencing of *Drosophila* spp. and theirnatural food sources, and transcriptomics of Drosophila melanogasterlarvae and *Drosophila*-associated yeasts | https://ddbj.nig.ac.jp/resource/bioproject/PRJDB15711 | DDBJ, PRJDB15711 |

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

# Appendix 1

## Appendix 1—key resources table

| Reagent type (species) or resource | Designation | Source or reference | Identifiers | Additional information |
|---|---|---|---|---|
| Genetic reagent (*Drosophila melanogaster*) | *Canton-Special* | EHIME-Fly *Drosophila* Stocks of Ehime University | E-10002 | |
| Genetic reagent (*D. melanogaster*) | *Cg-GAL4* | *Asha et al., 2003* (doi: 10.1093/genetics/163.1.203) | | |
| Strain, strain background (*Saccharomyces cerevisiae*) | BY4741 | NBRP-Yeast | BY23849 | |
| Strain, strain background (*Lactiplantibacillus plantarum*) | WJL | *Kim et al., 2021* (doi: 10.1038/s41586-021-03522-2) | | |
| Strain, strain background (*Lactiplantibacillus plantarum*) | *LactoBCAA* | *Kim et al., 2021* (doi: 10.1038/s41586-021-03522-2) | | |
| Sequence-based reagent | ITS3 | *White et al., 1990* | PCR primers | GCATCGATGAAGAACGCAGC |
| Sequence-based reagent | ITS4 | *White et al., 1990* | PCR primers | TCCTCCGCTTATTGATATGC |
| Sequence-based reagent | 341 F | *Hugerth et al., 2014* (doi: 10.1128/AEM.01403-14) | PCR primers | CCTACGGGNGGCWGCAG |
| Sequence-based reagent | 805R | *Hugerth et al., 2014* (doi: 10.1128/AEM.01403-14) | PCR primers | GACTACHVGGGTATCTAATCC |
| Sequence-based reagent | 518F | https://www.macrogen-japan.co.jp/cap_seq_0104.php | PCR primers | CCAGCAGCCGCGGTAATACG |
| Sequence-based reagent | NL1 | *Chandler et al., 2012* (doi: 10.1128/AEM.01741-12) | PCR primers | GCATATCAATAAGCGGAGGAAAAG |
| Sequence-based reagent | NL4 | *Chandler et al., 2012* (doi: 10.1128/AEM.01741-12) | PCR primers | GGTCCGTGTTTCAAGACGG |
| Sequence-based reagent | 27F | *Chandler et al., 2011* (doi: 10.1371/journal.pgen.1002272) | PCR primers | AGAGTTTGATCCTGGCTCAG |
| Sequence-based reagent | 1492R | *Chandler et al., 2011* (doi: 10.1371/journal.pgen.1002272) | PCR primers | GGTTACCTTGTTACGACTT |
| Commercial assay or kit | QIAamp DNA Stool Mini Kit | QIAGEN | Cat# 51504 | |
| Commercial assay or kit | THUNDERBIRD SYBR qPCR Mix | TOYOBO | Cat# QPS-201 | |
| Commercial assay or kit | InstaGene Matrix | Bio-Rad | Cat# 7326030 | |
| Commercial assay or kit | KOD FX | TOYOBO | Cat# KFX-101 | |
| Commercial assay or kit | Wizard SV Gel and PCR Clean-Up System | Promega | Cat# A9282 | |
| Commercial assay or kit | QIAGEN Genomic-tip 20G kit | QIAGEN | Cat# 10223 | |

*Appendix 1 Continued on next page*

*Appendix 1 Continued*

| Reagent type (species) or resource | Designation | Source or reference | Identifiers | Additional information |
|---|---|---|---|---|
| Commercial assay or kit | QIAseq FX DNA Library Kit | QIAGEN | Cat# 180477 | |
| Commercial assay or kit | Rapid Barcoding Kit | ONT | Cat# SQK-RBK004 | |
| Commercial assay or kit | RNeasy mini kit | QIAGEN | Cat# 74104 | |
| Commercial assay or kit | NEBNext Ultra II Directional RNA Library Prep Kit for Illumina | NEB | Cat# E7760 | |
| Commercial assay or kit | Biomasher II homogenizer kit | NIP | Cat# 320103 | |
| Chemical compound, drug | MRS Broth | Merck Millipore | Cat# 110661 | |
| Chemical compound, drug | YPD medium | Clontech | Cat# 630409 | |
| Chemical compound, drug | Potato Dextrose Broth | Sigma-Aldrich | Cat# P6685-250G | |
| Chemical compound, drug | Propionic acid | Nacalai Tesque | Cat# 29018-55 | |
| Chemical compound, drug | Cycloheximide | Wako | Cat# 037-20991 | |
| Chemical compound, drug | Ampicillin | Nacalai Tesque | | |
| Chemical compound, drug | Zymolyase-20T | Nacalai Tesque | Cat# 07663-91 | |
| Chemical compound, drug | Leucine | Nacalai Tesque | Cat# 20327-62 | |
| Chemical compound, drug | Isoleucine | Peptide Institute | Cat# 2712 | |
| Chemical compound, drug | Phloxine B | Wako | Cat# 166-02072 | |
| Software, algorithm | FLASH | *Magoč and Salzberg, 2011* (doi: 10.1093/bioinformatics/btr507) | | |
| Software, algorithm | CD-HIT-OTU | *Fu et al., 2012* (doi: 10.1093/bioinformatics/bts565); *Li and Godzik, 2006* (doi: 10.1093/bioinformatics/btl158) | | |
| Software, algorithm | QIIME | *Caporaso et al., 2010* (doi: 10.1038/nmeth.f.303) | | |
| Software, algorithm | NCBI BLAST search | *Johnson et al., 2008* (doi: 10.1093/nar/gkn201) | | |
| Software, algorithm | RDP sequence match tool | *Cole et al., 2014* (doi: 10.1093/nar/gkt1244) | | |
| Software, algorithm | Platanus_trim | http://platanus.bio.titech.ac.jp/pltanus_trim | | |
| Software, algorithm | GenomeScope | *Vurture et al., 2017* (doi: 10.1093/bioinformatics/btx153) | | |
| Software, algorithm | Albacore ver. 2.3.3. | ONT | | |

*Appendix 1 Continued on next page*

*Appendix 1 Continued*

| Reagent type (species) or resource | Designation | Source or reference | Identifiers | Additional information |
|---|---|---|---|---|
| Software, algorithm | NanoFilt | *De Coster et al., 2018* (doi: 10.1093/bioinformatics/bty149) | | |
| Software, algorithm | Porechop | https://github.com/rrwick/Porechop | | |
| Software, algorithm | MaSuRCA v3.2.6 | *Zimin et al., 2013* (doi: 10.1093/bioinformatics/btt476) | | |
| Software, algorithm | BUSCO | *Simão et al., 2015* (doi: 10.1093/bioinformatics/btv351) | | |
| Software, algorithm | GenomeMatcher | *Ohtsubo et al., 2008* (doi: 10.1186/1471-2105-9-376) | | |
| Software, algorithm | FunGAP | *Min et al., 2017* (doi: 10.1093/bioinformatics/btx353) | | |
| Software, algorithm | Hisat2 | *Kim et al., 2015* (doi: 10.1038/nmeth.3317) | | |
| Software, algorithm | gplots | *Warnes et al., 2022* (https://CRAN.R-project.org/package=gplots) | | |
| Software, algorithm | R | *R Development Core Team, 2020* | RRID:SCR_001905 | |
| Software, algorithm | Database for annotation, visualization and integrated discovery (DAVID) Functional Annotation Chart | *Huang et al., 2009* (doi: 10.1038/nprot.2008.211); *Sherman et al., 2022* (doi: 10.1093/nar/gkac194) | | |
| Software, algorithm | MetaboAnalystR 3.3.0. | *Pang et al., 2020* (doi: 10.3390/metabo10050186) | | |
| Other | UltrafreeMC-PLHCC for Metabolome Analysis column | Human Metabolome Technologies | Cat# UFC3LCCNB-HMT | See 'Preparing samples for LC–MS analysis' in Materials and methods |
| Other | Discovery HS F5-3 column | Sigma-Aldrich | Cat# 567503-U | See 'LC–MS/MS measurement' in Materials and methods |

