## [Editor Report · eLife assessment]

This is an **important** study that addresses a significant question in microbiome research. The authors provide **convincing** evidence that certain bacterial groups within the fly microbiome have critical functions for host development. Additionally, dietary aspects such as microbial community progression in a natural food source are integrated into their host-microbe interaction analyses.

---

## [Referee Report · Reviewer #1 (Public Review)]

Summary:

This valuable study analyzes the contribution of fungal and bacterial microbiota species to the growth and development of *Drosophila*. The authors use bacterial and fungal species associated with *Drosophila* in the wild to analyze their respective contributions in promoting larval growth in a decaying banana, mimicking the natural niche of fruit fly. They found that some fungal species and some fungus/bacteria combinations effectively promote growth by supplementing key branched amino acids in the food substratum. Production of these amino acids by *Drosophila* itself is not sufficient, and only fungal species that secrete these amino acids into the medium can sustain *Drosophila* growth. Thus, the authors clarify how facultative symbionts contribute to host growth in a natural setting by changing the food substratum in a dynamic manner.

Strengths:

The natural setting developed by the authors to analyze the impact of the microbiota is clearly valuable, as is the focus on the role of fungal microbiota species. This complements studies of *Drosophila* microbiota that have previously focused on bacterial species and used a lab setting.

While there has been an extensive focus on obligate endosymbionts or gut symbionts, this study analyzes how facultative symbionts shape the food substratum and influence host growth.

A last strength of this study is that it analyzes the contribution of *Drosophila* microbiota over a dynamic timeframe, analyzing how microbial species change in decaying fruit over time.

Weaknesses:

1. The author should better review what we know of fungal *Drosophila* microbiota species as well as the ecology of rotting fruit. Are the microbiota species described in this article specific to their location/setting? It would have been interesting to know if similar species can be retrieved in other locations using other decaying fruits. The term 'core' in the title suggests that these species are generally found associated with *Drosophila* but this is not demonstrated. The paper is written in a way that implies the microbiota members they have found are universal. What is the evidence for this? Have the fungal species described in this paper been found in other studies? Even if this is not the case, the paper is interesting, but there should be a discussion of how generalizable the findings are.

2. Can the author clearly demonstrate that the microbiota species that develop in the banana trap are derived from flies? Are these species found in flies in the wild? Did the authors check that the flies belong to the *D. melanogaster* species and not to the sister group D. simulans?

3. Did the microarrays highlight a change in immune genes (ex. antibacterial peptide genes)? Whatever the answer, this would be worth mentioning. The authors described their microarray data in terms of fed/starved in relation to the Finke article. This is fine they should clarify if they observed significant differences between species (differences between species within bacteria or fungi, and more generally differences between bacteria versus fungi).

4. The whole paper - and this is one of its merits - points to a role of the *Drosophila* larval microbiota in processing the fly food. Are these bacterial and fungal species found in the gut of larvae/adults? Are these species capable of establishing a niche in the cardia of adults as shown recently in by the Ludington lab (Dodge et al.,)? Previous studies have suggested that microbiota members stimulate the Imd pathway leading to an increase in digestive proteases (Erkosar/Leulier). Are the microbiota species studied here affecting gut signaling pathways beyond providing branched amino acids?

---

## [Referee Report · Reviewer #2 (Public Review)]

Summary:

In this manuscript, Mure et al investigated host-microbe interactions in wild-mimicked settings. They analyzed microbiome composition using bananas that had been fed on by wild larvae and found that the microbiota composition shifted from the early stage of feeding to the later stage of the fermentation process proceeded. They isolated several yeast and bacterial species from the food, and examined larval growth on banana-based food, mimicking natural setting where germ-free larvae cannot grow on it. The authors found that a yeast, Hanseniaspora uvarum, can support larval growth sufficiently, and insists that branched-chain amino acids (BCAAs) provided by the yeast may partly be accounted for the growth support. Interestingly, other isolated yeast species, some were non-supportive strains in terms of larval growth, can assist larval development when they were heat-killed. Besides, they showed that acetic acid bacteria, isolated from well-fermented banana (later-stage food), is sufficiently supportive but their presence depended on other microbes, lactic acid bacteria or yeast.

Strengths:

So far, host-microbe studies using *Drosophila melanogaster* have relatively less focused on the roles of fungi and many studies used only "model" yeasts. In the experimental setting where natural conditions may be well mimicked, the authors successfully isolated wild yeast species and convincingly showed that wild yeast plays a critical role in promoting host growth. In addition, the authors provided intriguing observations that all of the heat-killed yeast promoted larval growth even though some of the yeast never support the development when they were alive, suggesting that wild yeasts produce the necessary nutrients for larval development, but the nutrients of non-supportive yeasts are not accessible to the host. This might be an interesting indication for further studies revealing host-fungi interactions.

Weaknesses:

The experimental setting that, the authors think, reflects host-microbe interactions in nature is one of the key points. However, it is not explicitly mentioned whether isolated microbes are indeed colonized in wild larvae of *Drosophila melanogaster* who eat bananas. Another matter is that this work is rather descriptive. A molecular level explanation is missing in "interspecies interactions" between lactic acid bacteria (or yeast) and acetic acid bacteria that assure their inhabitation.

---

## [Referee Report · Reviewer #3 (Public Review)]

Summary: In this manuscript, Mure et al. describe interactions between diet, microbiome, and host development using *Drosophila* as a model. By characterizing microbial communities in food sources and animals, the authors showed that microbial community dynamics in the food source is critical for host development.

Strengths: This is a very interesting study where authors managed to tackle a difficult question in an elegant manner. How the interactions between different microbial species within the microbiome shape host physiology is an area of great interest but equally challenging due to the complexity of intercellular interactions in complex, host-associated microbial communities. By using a simplified model and interrogating not only microbe-microbe and host-microbe interactions, but also the role played by diet, authors were able to identify significant interactions during fly development.

Weaknesses: All weaknesses observed in the original manuscript have been corrected in the current version.

---

## [Author Response]

The following is the authors’ response to the original reviews.

We would like to thank the reviewers for their thoughtful evaluation of our manuscript. We considered all the comments and prepared the revised version. The following are our responses to the reviewers’ comments. All references, including those in the original manuscript are included at the end of this point-by-point response.

**Reviewer #1 (Public Review):**
Weaknesses:1. The authors should better review what we know of fungal *Drosophila* microbiota species as well as the ecology of rotting fruit. Are the microbiota species described in this article specific to their location/setting? It would have been interesting to know if similar species can be retrieved in other locations using other decaying fruits. The term 'core' in the title suggests that these species are generally found associated with *Drosophila* but this is not demonstrated. The paper is written in a way that implies the microbiota members they have found are universal. What is the evidence for this? Have the fungal species described in this paper been found in other studies? Even if this is not the case, the paper is interesting, but there should be a discussion of how generalizable the findings are.

The reviewer inquires as to whether the microbial species described in this article are ubiquitously associated with *Drosophila* or not. Indeed, most of the microbes described in this manuscript are generally recognized as species associated with *Drosophila* spp. For example, yeasts such as Hanseniaspora uvarum, Pichia kluyveri, and Starmerella bacillaris have been detected in or isolated from *Drosophila* spp. collected in European countries as well as the United States and Oceania (Chandler et al., 2012; Solomon et al., 2019). As for bacteria, species belonging to the genera Pantoea, Lactobacillus, Leuconostoc, and Acetobacter have also previously been detected in wild *Drosophila* spp. (Chandler et al.,2011). These statements have been incorporated into our revised manuscript (lines 391-397).Nevertheless, the term “core” in the manuscript and title may lead tomisunderstanding, as the generality does not ensure the ubiquitous presence of these microbial species in every individual fly. Considering this point, we replaced the “core” with “key,” a term that is more appropriate to our context.

1. Can the authors clearly demonstrate that the microbiota species that develop in the banana trap are derived from flies? Are these species found in flies in the wild? Did the authors check that the flies belong to the *D. melanogaster* species and not to the sister group D. simulans?Can the authors clearly demonstrate that the microbiota species that develop in the banana trap are derived from flies? Are these species found in flies in the wild?

The reviewer asked whether the microbial species detected from the fermented banana samples were derived from flies. To address this question, additional experiments under more controlled conditions would be needed, such as artificially introducing wild flies onto fresh bananas in the laboratory. Nevertheless, the microbes potentially originate from wild flies, as supported by the literature cited in our response to the Weakness 1.

Alternative sources of microbes also merit consideration. For example, microbes may have been introduced to unfermented bananas by penetration through peel injuries (lines 1300-1301). In addition, they could be introduced by insects other than flies, given that rove beetles (Staphylinidae) and sap beetles (Nitidulidae) were observed in some of the traps.The explanation of these possibilities have been incorporated into DISCUSSION (lines 414427) of our revised manuscript.

Did the authors check that the flies belong to the *D. melanogaster* species and not to the sister group D. simulans?

Our sampling strategy was designed to target not only *D. melanogaster* but also other domestic *Drosophila* species, such as D. simulans, that inhabit human residential areas. For the traps where adult flies were caught, we identified the species of the drosophilids as shown in Table S1, thereby showing the presence of either or both *D. melanogaster* and D. simulans. We added these descriptions in MATERIALS AND METHODS (lines 511-512 and 560-562), and DISCUSSION (lines 378-379).

1. Did the microarrays highlight a change in immune genes (ex. antibacterial peptide genes)? Whatever the answer, this would be worth mentioning. The authors described their microarray data in terms of fed/starved in relation to the Finke article. They should clarify if they observed significant differences between species (differences between species within bacteria or fungi, and more generally differences between bacteria versus fungi).Did the microarrays highlight a change in immune genes (ex. antibacterial peptide genes)?Whatever the answer, this would be worth mentioning.

Regarding the antimicrobial peptide genes, statistical comparisons of our RNA-seq data across different conditions were impracticable because most of the genes showed low expression levels. The RNA-seq data of the yeast-fed larvae is shown in Author response Table 1. While a subset of genes exhibited significantly elevated expression in the nonsupportive conditions relative to the supportive ones, this can be due to intra-sample variability rather than the difference in the nutritional conditions. Similar expression profiles were observed in the bacteria-fed larvae as well (data not shown). Therefore, it is difficult to discuss a change in immune genes in the paper. Additionally, the previous study that conducted larval microarray analysis (Zinke et al., 2002) did not explicitly focus on immune genes.

**Author response table 1. sa4table1:** Antimicrobial peptide genes are not up-regulated by any of the microbes. Antimicrobial peptides gene expression profiles of whole bodies of first-instar larvae fed on yeasts. TPM values of all samples and comparison results of gene expression levels in the larvae fed on supportive and non-supportive yeasts are shown. Antibacterial peptide genes mentioned in Hanson and Lemaitre, 2020 are listed. NA or na, not available.

	TRM mulem																													
				M sabites				avasat		arm-tios																				
						a	2	a^(3)+2x-2								2 =								N.wa,os sin s				N.ws,a twac		
Anososen	opt	0													0	0	4	0				51	na	m	mas	m	ma	NA	NA	Ma
	pos				8			g				g			if		A3	g.	6	8		昰	na	m	m	m	m	AA	MA	m
Egno1 1200	sh	0			0			0				2			0	01	11	0	6	0		6	na	m	m	ma		MA	MA	
	As	0	0		0			0	0	0		5	0		0	02	26	0	0	0		15	na	m	mo	m	man	AA	NA	
tan oo415	sac	0			0			a	0	0		a			a	02		0	a			a	nà	m	m	m	m	NA	NA	ma
	Mo	0	0		o			G		I		G			g	需	g	0				o	na	m	ma	m	ma	MA	MA	Ma
tan 01039	oo									a		3	0		6		10	0				7	na	m	in			AA	MA	ma
	sec				0			6				1			0	0	4	0	0		0	15	na		ma	as	m	NA	MA	in
	oacses	0			0			可	0	0		可			0	6	1	0	0		a	百	na	ma	mis	m	ma	NA	NA	m
	aces	0			a			百	0	0		a			of	0	0	0	0	o	0	8	na	m	in	in	m	NA	NA	ma
	cos							g	g	if		1)	a		0	8	1	0.			8	4	na	m						
Fono00034	anp	0			of	0			0	pi		of			0	6	0	0	8		0	0	na	m	ma	na	ma	AA	AA	44
	det	0			0			0				0			3	0	0	0	0	0	0	1	na	m	m	ma		MA	NA	mas
	a_(5)							可	1		3	8	0		o.	18	25	a	5	13	8	0.	ná							mas
	axsin	0						G				G			g	0	g	0			a	0	na	成	(日)	用	m	MA	MA	m
	|x:22				8		0	a		o		a			8	a	a	0			a	a	na	m	it	II	m	MA	MA	Ma
Egnos	orel	0			0			0	0	0		0			0	0	0	0	0		0	0	na	ma	ma	ma	m	NA	NA	14
	ares	0			0		0		6	0		0			theta	0	0	0		0	0	0	na	m	m	ma	in	NA	NA	in
00035444	ars 5	a			1			a	4	a		a			0	0	0	1	2		0	6	na	ma	in	m	m	NA	NA	
	axse	0			of			g	g			G				g	g	0	6		rarr	0	na	m	m	m	ma	MA	MA	m
	m				2		2		2	0		5			2	25		1			2	2	n						NA	m
国onoses	min	5					(4)/(4)		3	0	6	可	(1)/(1)			4	23	1	1	3	1	2	ns	pi	ma	ma	0.28	0.16	NA	ins
	M:4	1						百	a	0		讨		G	a	06	可	6	0	0	if	12	na	m	m	กล	m	NA	NA	Nas
Æoncos4512		3							10	11		8	7	(3)	15	172	27	9	13			23	M∼ooAF=N			K minsin x	-1.54	-1.97	-1.o+	
tonosys9	Mt	0	0		0						19				1	211	16	1			1	3	na		ia	m	M	NA	NA	m
	कotat &	1			1							a			11	13.	16	2			10	1a	na	n=0sin s	mi	K sinsise	NA	-200	NA	
Enos:	Me	0			0			者			pi				0	8	66	0			10	4	na	sin	ros	m	sh	NA	NA	m
	mos	0			8			保	g	8	12				rarr		01	0		8	2	pi	na	ma	-	m	MA	MA	NA	m
	orsin:	2		8	2							a)			1	22	24	3	3	15	11		na	ma	m		m	MA	MA	ma
	cosse:	0						0							13	213	36	0			14:	a	กล	r sin,< 5sax	a		ma	908	NA	-925
	03.420	0													8	0	1	6			6	1	na		is	m	Na	NA	NA	Ma
		3			theta			0							0	0	5	0				8	na	na	ai	m	M	NA	NA	mh
Enos	cas7a	0						a							2	1	3					-		m						Ma
	mes	0													0	theta1	10	0			G	6	na	in	m	m	res	13	A	mas
tonow3	काळब	0	0		f	可			0	g	1	图			0	11	19	0		0		3	na	is	m	ma	ma	NA	NA	ma
		0	0		0													0					na	กิ		ma	ma	NA	NA	MA

They should clarify if they observed significant differences between species (differences between species within bacteria or fungi, and more generally differences between bacteria versus fungi).

We did not observe significant differences in the gene expression profiles of the larvae fed on different microbial species within bacteria or fungi, or between those fed on bacteria and those fed on fungi. For example, the gene expression profiles of larvae fed on the various supportive microbes showed striking similarities to each other, as evidenced by the heat map showing the expression of all genes detected in larvae fed either yeast or bacteria (Author response image 1). Similarities were also observed among larvae fed on various nonsupportive microbes.

Only a handful of genes showed different expression patterns between larvae fed on yeast and those fed on bacteria. Thus, it is challenging to discuss the potential differential impacts of yeast and bacteria on larval growth, if any.

**Author response image 1. sa4fig1:** Gene expression profiles of larvae fed on the various supporting microbes show striking similarities to each other. Heat map showing the gene expression of the first-instar larvae that fed on yeasts or bacteria. Freshly hatched germ-free larvae were placed on banana agar inoculated with each microbe and collected after 15 h feeding to examine gene expression of the whole body. Note that data presented in Figures 3A and 4C in the original manuscript, which are obtained independently, are combined to generate this heat map. The labels under the heat map indicate the microbial species fed to the larvae, with three samples analyzed for each condition. The lactic acid bacteria (“LAB”) include Lactiplantibacillus plantarum and Leuconostoc mesenteroides, while the lactic acid bacterium (“AAB”) represents Acetobacter orientalis. “LAB + AAB” signifies mixtures of the AAB and either one of the LAB species. The asterisks in the label highlight “LAB + AAB” or “LAB” samples clustered separately from the other samples in those conditions; “*” indicates a sample in a “LAB + AAB” condition (Lactiplantibacillus plantarum + Acetobacter orientalis), and “**” indicates a sample in a “LAB” condition (Leuconostoc mesenteroides). Brown abbreviations of scientific names are for the yeast-fed conditions. H. uva, Hanseniaspora uvarum; K. hum, Kazachstania humilis; M. asi, Martiniozyma asiatica; Sa. cra, Saccharomycopsis crataegensis; P. klu, Pichia kluyveri; St.bac, Starmerella bacillaris; BY4741, *Saccharomyces cerevisiae* BY4741 strain.

1. The whole paper - and this is one of its merits - points to a role of the *Drosophila* larval microbiota in processing the fly food. Are these bacterial and fungal species found in the gut of larvae/adults? Are these species capable of establishing a niche in the cardia of adults as shown recently in the Ludington lab (Dodge et al.,)? Previous studies have suggested that microbiota members stimulate the Imd pathway leading to an increase in digestive proteases (Erkosar/Leulier). Are the microbiota species studied here affecting gut signaling pathways beyond providing branched amino acids?The whole paper - and this is one of its merits - points to a role of the *Drosophila* larval microbiota in processing the fly food. Are these bacterial and fungal species found in the gut of larvae/adults? Are these species capable of establishing a niche in the cardia of adults as shown recently in the Ludington lab (Dodge et al.,)?

Although we did not investigate the microbiota in the gut of either larvae or adults, we did compare the microbiota within surface-sterilized larvae or adults with the microbiota in food samples. We found that adult flies and early-stage foods, as well as larvae and late-stage foods, harbored similar microbial species (Figure 1F). Additionally, previous studies examining the gut microbiota in wild adult flies have detected microbes belonging to the same species or taxa as those isolated from our foods (Chandler et al., 2011; Chandler et al., 2012). We have elaborated on this in our response to Weakness 1.

While we did not investigate whether these species are capable of establishing a niche in the cardia of adults, we have cited the study by Dodge et al., 2023 in our revised manuscript and discussed the possibility that predominant microbes in adult flies may show a propensity for colonization (lines 410-413).

Previous studies have suggested that microbiota members stimulate the Imd pathway leading to an increase in digestive proteases (Erkosar/Leulier). Are the microbiota species studied here affecting gut signaling pathways beyond providing branched amino acids?

The reviewer inquires whether the supportive microbes in our study stimulate gut signaling pathways and induce the expression of digestive protease genes, as demonstrated in a previous study (Erkosar et al., 2015). Based on our RNA-seq data, this is unlikely. The aforementioned study demonstrated that seven protease genes are upregulated through Imd pathway stimulation by a bacterium that promotes the larval growth. In our RNA-seq analysis, these seven genes did not exhibit a consistent upregulation in the presence of the supportive microbes (H. uva or K. hum in Author response table 2A; Le. mes + A. ori in Author response table 2B). Rather, they exhibited a tendency to be upregulated by the presence of non-supportive microbes (St. bac or Pi. klu in Author response table 2A; La. pla in Author Response Table 2B).

**Author response table 2. sa4table2:** Most of the peptidase genes reported by Erkosar et al. , 2015 are more highly expressed under the non-supportive conditions than the supportive conditions. Comparison of the expression levels of seven peptidase genes derived from the RNA-seq analysis of yeast-fed (A) or bacteria-fed (B) first-instar larvae. A previous report demonstrated that the expression of these genes is upregulated upon association with a strain of Lactiplantibacillus plantarum, and that the PGRP-LE/Imd/Relish signaling pathway, at least partially, mediates the induction (Erkosar et al., 2015). H. uva, Hanseniaspora uvarum; K. hum, Kazachstania humilis; P. klu, Pichia kluyveri; S. bac, Starmerella bacillaris; La. pla, Lactiplantibacillus plantarum; Le. mes, Leuconostoc mesenteroides; A. ori, Acetobacter orientalis; ns, not significant.

FlyBase ID	symbol	gene_name	Significant difference between conditions				log 2 (fold change)			
			H. uva (H)or P. klu (P)	H. uva (H)or S. bac (S)	K. hum (K)or P. Kiu (P)	K. hum (K)or S. bac (S)	H. uva/P. Kiu	H. uva/S. bac	K. hum/P. klu	K. humIS. bac
FBgn0035887	Jon 66 Cii	Jonah 66Cii	H > P	H > S	ns	ns	0.47	0.50	-0.24	-0.21
FBgn0035886	Jon 66Ci	Jonah 66Ci	H <= P	H≪S	K <= P		-1.27	-0.91	-1.31	-0.94
FBgn0001285	Jon44E	Jonah 44E	H < P	H < S	K < P	K&S	-0.66	-0.52	-0.96	-0.82
FBgn0035667	Jon65Ai	Jonah 65Ai	H < P	H < S	K < P	K < S	-0.51	-0.59	-0.60	-0.68
FBgn0003358	Jon 99Ci	Jonah 99Ci	H > P	H > S		Kos	0.77	0.97	0.67	0.87
FBgn0036023	CG18179	-	H < P	H < S	K < P	K < S	-1.38	-0.55	-1.38	-0.56
FBgn0036024	CG18180	-	H < P	H < S	K < P		-0.84	-0.69	-0.53	-0.38

**Reviewer #2 (Public Review):**
Weaknesses:The experimental setting that, the authors think, reflects host-microbe interactions in nature is one of the key points. However, it is not explicitly mentioned whether isolated microbes are indeed colonized in wild larvae of *Drosophila melanogaster* who eat bananas. Another matter is that this work is rather descriptive and a few mechanical insights are presented. The evidence that the nutritional role of BCAAs is incomplete, and molecular level explanation is missing in "interspecies interactions" between lactic acid bacteria (or yeast) and acetic acid bacteria that assure their inhabitation. Apart from these matters, the future directions or significance of this work could be discussed more in the manuscript.

The experimental setting that, the authors think, reflects host-microbe interactions in nature is one of the key points. However, it is not explicitly mentioned whether isolated microbes are indeed colonized in wild larvae of *Drosophila melanogaster* who eat bananas.

The reviewer asks whether the isolated microbes were colonized in the larval gut. Previous studies on microbial colonization associated with *Drosophila* have predominantly focused on adults (Pais et al. PLOS Biology, 2018), rather than larval stages. Developing larvae continually consume substrates which are already subjected to microbial fermentation and abundant in live microbes until the end of the feeding larval stage. Therefore, we consider it difficult to discuss microbial colonization in the larval gut. We have mentioned this point in DISCUSSION of the revised manuscript (lines 408-410).

Another matter is that this work is rather descriptive and a few mechanical insights are presented. The evidence that the nutritional role of BCAAs is incomplete, and molecular level explanation is missing in "interspecies interactions" between lactic acid bacteria (or yeast) and acetic acid bacteria that assure their inhabitation.While we recognize the importance of comprehensive mechanistic analysis, elucidation of more detailed molecular mechanisms lies beyond the scope of this study and will be a subject of future research.

Regarding the nutritional role of BCAAs, the incorporation of BCAAs enabled larvae fed with the non-supportive yeast to grow to the second-instar stage. This observation implies that consumption of BCAAs upregulates diverse genes involved in cellular growth processes in larvae. We mentioned a previously reported interaction between lactic acid bacteria (LAB) and acetic acid bacteria (AAB) in the manuscript (lines 433-436). LAB may facilitate lactate provision to AAB, consequently enhancing the biosynthesis of essential nutrients such as amino acids. To test this hypothesis, future experiments will include the supplementation of lactic acid to AAB culture plates, and the co-inoculation of AAB with LAB mutant strains defective in lactate production to assess both larval growth and continuous larval association with AAB. With respect to AAB-yeast interactions, metabolites released from yeast cells might benefit AAB growth, and this possibility will be investigated through the supplementation of AAB culture plates with candidate metabolites identified in the cell suspension supernatants of the late-stage yeasts.

Apart from these matters, the future directions or significance of this work could be discussed more in the manuscript.

We appreciate the reviewer's recommendations. The explanation of the universality of our findings has been included in the revised DISCUSSION (lines 391-397). We have also added descriptions on the implication of compositional shifts occurring in adult microbiota (lines 404413), possible inoculation routes of different microbes (lines 414-427), and hypotheses on the mechanism of larval growth promotion by yeasts (lines 469-476), all of which could be the focus of our future study.

**Reviewer #3 (Public Review):**
Weaknesses:Despite describing important findings, I believe that a more thorough explanation of the experimental setup and the steps expected to occur in the exposed diet over time, starting with natural "inoculation" could help the reader, in particular the non-specialist, grasp the rationale and main findings of the manuscript. When exactly was the decision to collect earlystage samples made? Was it when embryos were detected in some of the samples? What are the implications of bacterial presence in the no-fly traps? These samples also harbored complex microbial communities, as revealed by sequencing. Were these samples colonized by microbes deposited with air currents? Were they the result of flies that touched the material but did not lay eggs? Could the traps have been visited by other insects? Another interesting observation that could be better discussed is the fact that adult flies showed a microbiome that more closely resembles that of the early-stage diet, whereas larvae have a more late-stage-like microbiome. It is easy to understand why the microbiome of the larvae would resemble that of the late-stage foods, but what about the adult microbiome? Authors should discuss or at least acknowledge the fact that there must be a microbiome shift once adults leave their food source. Lastly, the authors should provide more details about the metabolomics experiments. For instance, how were peaks assigned to leucine/isoleucine (as well as other compounds)? Were both retention times and MS2 spectra always used? Were standard curves produced? Were internal, deuterated controls used?When exactly was the decision to collect early-stage samples made? Was it when embryos were detected in some of the samples?

We collected traps and early-stage samples 2.5 days after setting up the traps. This duration was determined from pilot experiments. A shorter collection time resulted in a lower likelihood of obtaining traps visited by adult flies, whereas a longer collection time caused overcrowding of larvae as well as deaths of adults from drowning in the liquid seeping out of the fruits. These procedural details have been included in the MATERIALS AND METHODS section of the revised manuscript (lines 523-526).

What are the implications of bacterial presence in the no-fly traps? These samples also harbored complex microbial communities, as revealed by sequencing. Were these samples colonized by microbes deposited with air currents? Were they the result of flies that touched the material but did not lay eggs? Could the traps have been visited by other insects?

We assume that the origins of the microbes detected in the no-fly trap foods vary depending on the species. For instance, Colletotrichum musae, the fungus that causes banana anthracnose, may have been present in fresh bananas before trap placement. The filamentous fungi could have originated from airborne spores, but they could also have been introduced by insects that feed on these fungi. We have included these possibilities in the DISCUSSION section of the revised manuscript (lines 417-421).

Another interesting observation that could be better discussed is the fact that adult flies showed a microbiome that more closely resembles that of the early-stage diet, whereas larvae have a more late-stage-like microbiome. It is easy to understand why the microbiome of the larvae would resemble that of the late-stage foods, but what about the adult microbiome? Authors should discuss or at least acknowledge the fact that there must be a microbiome shift once adults leave their food source.

We are grateful for the reviewer's insightful suggestion regarding shifts in the adult microbiome. We have included in the DISCUSSION section of the revised manuscript the possibility that the microbial composition may change substantially during pupal stages or after adult eclosion (lines 404-413).

Lastly, the authors should provide more details about the metabolomics experiments. For instance, how were peaks assigned to leucine/isoleucine (as well as other compounds)?Were both retention times and MS2 spectra always used?

In this metabolomic analysis, LC-MS/MS with triple quadrupole MS monitors the formation of fragment ions from precursor ions specific to each target compound. The use of PFPP columns, which provide excellent separation of amino acids and nucleobases, allows chromatographic peaks of many structural isomers to be separated into independent peaks. In addition, all measured compounds are compared with data from a standard library to confirm retention time agreement. Structural isomers were separated either by retention time on the column or by compound-specific MRM signals (in fact, leucine and isoleucine have both unique MRM channels and column separations). Detailed MRM conditions are identical to the previously published study (Oka et al., 2017). These have been included in the revised ‘LC-MS/MS measurement’ section in MATERIALS AND METHODS (lines 810-824).

Were standard curves produced?

Since relative quantification of metabolite amounts was performed in this study, no standard curve was generated to determine absolute concentrations. However, a standard compound of known concentration (single point) was measured to confirm retention time and relative area values.

Were internal, deuterated controls used?

Internal standards for deuterium-labeled compounds were not used in this study. This is because it is not realistic to obtain deuterium-labeled compounds for all compounds since a large number of compounds are measured. However, an internal standard (L-methionine sulfone) is added to the extraction solvent to calculate the recovery rate. This has been included in the revised ‘LC-MS/MS measurement’ section in MATERIALS AND METHODS (lines 824-825).

**Reviewer #1 (Recommendations For The Authors):**
Additional comments1. The authors should do a better job of presenting their data. It took me quite a while to understand the protocol of Figure 1. Panel 1A, B, C could be improved. For instance, 1A suggests that flies are transferred to the lab while this is in fact the banana trap. Indicate 'Banana trap colonized by flies' rather 'wild-type flies in the trap'. 1C: should indicate that the food suspension comes from the banana trap. 1B,D,D: do not use pale color as legend. Avoid the use of indices in Figure 2 (Y1 rather than Y1). Grey colors are difficult to distinguish in Figure 2. Etc. It is a pain for reviewers that figure legends are on the verso of each figure and not just below.

We thank the reviewer for the detailed suggestions to improve the clarity and comprehensibility of our figures. We have improved the figures according to the suggestions. As for the figure legends, we have placed them below each respective figure whenever possible.

1. Clarify in the text if 'sample' means food substratum or flies/larvae (ex. line 116 and elsewhere).

We have revised the word “sample” throughout our manuscript and eliminated the confusion.

1. Line 170 - clarify what you mean by fermented food.

We have replaced the “fermented larval foods” with “fermented bananas” in our revised manuscript (line 165).

1. Line 199 - what is the meaning of 'stocks'.

We have replaced the “stocks” with “strains” (line 195).

1. Line 320 - explain more clearly what the yeast-conditioned banana-agar plate and cell suspension supernatant are, and what the goals of using these media are. This will help in understanding the subsequent text.

We have added a supplemental figure illustrating the sample preparation for the metabolomic analysis (Figure S6), with the following legend describing the procedure (lines 1335-1346): “Sample preparation process for the metabolomic analysis. We suspected that the supportive live yeast cells may release critical nutrients for larval growth, whereas the non-supportive yeasts may not. To test this possibility, we made three distinct sample preparations of individual yeast strains (yeast cells, yeast-conditioned banana-agar plates, and cell suspension supernatants). Yeast cells were for the analysis of intracellular metabolites, whereas yeast-conditioned banana-agar plates and cell suspension supernatants were for that of extracellular metabolites. The samples were prepared as the following procedures. Yeasts were grown on banana-agar plates for 2 days at 25°C, and then scraped from the plates to obtain “yeast cells.” Next, the remaining yeasts on the resultant plates were thoroughly removed, and a portion from each plate was cut out (“yeast-conditioned banana agar”). In addition, we suspended yeast cells from the agar plates into sterile PBS, followed by centrifugation and filtration to eliminate the yeast cells, to prepare “cell suspension supernatants.”

1. Figure 5 is difficult to understand. Provide more explanation. Consider moving the 'all metabolites panel' to Supp. Better explain what this holidic medium is.

The holidic medium is a medium that has been commonly used in the *Drosophila* research community, which contains ~40 known nutrients, and supports the larval development to pupariation (Piper et al., 2014; Piper et al., 2017). We have introduced this explanation to the RESULTS section of the manuscript (lines 322-327). However, the scope of our research reaches beyond the analysis of the holidic medium components, because feeding the holidic medium alone causes a significant delay in larval growth, suggesting a lack of nutritional components (Piper et al., 2014). Thus, we believe the "All Metabolites" panels should be placed alongside the corresponding “The holidic medium components” panels.

1. I could not access Figure 6 when downloading the PDF. The page is white and an error message appears - it is problematic to review a paper lacking a figure.

We regret any inconvenience caused, perhaps due to a system error. Please refer to the Author response image 2, which is identical to Figure 6 of our original manuscript.

**Author response image 2. sa4fig2:** Supportive yeasts facilitate larval growth by providing nutrients, including branched-chain amino acids, by releasing them from their cells (Figure 6 from the original manuscript). (A and B) Growth of larvae feeding on yeasts on banana agar supplemented with leucine and isoleucine. (A) The mean percentage of the live/dead individuals in each developmental stage. n=4. (B) The percentage of larvae that developed into second instar or later stages. The “Not found” population in Figure 6A was omitted from the calculation. Each data point represents data from a single tube. Unique letters indicate significant differences between groups (Tukey-Kramer test, p < 0.05). (C) The biosynthetic pathways for leucine and isoleucine with *S. cerevisiae* gene names are shown. The colored dots indicate enzymes that are conserved in the six isolated species, while the white dots indicate those that are not conserved. Abbreviations of genera are given in the key in the upper right corner. LEU2 is deleted in BY4741. (D-G) Representative image of Phloxine B-stained yeasts. The right-side images are expanded images of the boxed areas. The scale bar represents 50 µm. (H) Summary of this study. H. uvarum is predominant in the early-stage food and provides Leu, Ile, and other nutrients that are required for larval growth. In the late-stage food, AAB directly provides nutrients, while LAB and yeasts indirectly contribute to larval growth by enabling the stable larva-AAB association. The host larva responds to the nutritional environment by dramatically altering gene expression profiles, which leads to growth and pupariation. H. uva, Hanseniaspora uvarum; K. hum, Kazachstania humilis; Pi. klu, Pichia kluyveri; St. bac, Starmerella bacillaris; GF, germ-free.

1. Line 323 - Consider rewriting this sentence (too long, explain what the holidic medium is and why this is interesting). "In the yeast-conditioned banana-agar plates, which were anticipated to contain yeast-derived nutrients, many well-known nutrients included in a chemically defined synthetic (holidic) medium for *Drosophila melanogaster* (Piper et al., 2014, 2017) were not increased compared to the sterile banana-agar plates; instead, they exhibited drastic decreases irrespective of the yeast species."

We thank the reviewer's suggestion to improve the readability of our manuscript. We have rewritten the sentence in the revised manuscript (lines 320-328) as follows: “The yeastconditioned banana-agar plates were expected to contain yeast-derived nutrients. On the contrary, the result revealed a depletion of various metabolites originally present in the sterile banana agar (Figure 5A). This result prompted us to focus on the metabolites in the chemically defined (holidic) medium for *Drosophila melanogaster* Piper et al., 2014; Piper et al., 2017. This medium contains ~40 known nutrients, and supports the larval development to pupariation, albeit at the half rate compared to that on a yeast-containing standard laboratory food Piper et al., 2014; Piper et al., 2017. Therefore, the holidic medium could be considered to contain the minimal essential nutrients required for larval growth. Our analysis indicated a substantial reduction of these known nutrients in the yeast-conditioned plates compared to their original quantities (Figure 5B).”

**Reviewer #2 (Recommendations For The Authors):**
Suggestions for improved or additional experiments, data or analyses.1. It should be clearly shown (or stated) that isolated microbes, such as H. uvarum and Pa. agglomerans, are indigenous microbes in wild *Drosophila melanogaster* in their outdoor sampling.

We thank the reviewer for the suggestions. Addressing the presence of isolated microbes within wild *D. melanogaster* adults is important, but cannot be feasible with our data for the following reasons. Our microbiota analysis of adults was conducted using pooled individuals of multiple *Drosophila* species, rather than using *D. melanogaster* exclusively. Moreover, the microbial isolation and the analysis of adult microbiota were carried out in two independent samplings (Figures 1A and 1E in the original manuscript, respectively). As a result, the microbial species detected in the adults were slightly different from those isolated from the food samples collected in the previous sampling. Nevertheless, it is worth noting that H. uvarum dominated in 2 out of the 3 adult samples, constituting >80% of the fungal composition. Pantoea agglomerans was not detected in the adults, although Enterobacterales accounted for >59% in 2 out of the 3 samples. Therefore, these isolated microbial species, or at least their phylogenetically related species, are presumed to be indigenous to wild *D. melanogaster*.

If the reviewer’s suggestion was to state the dominance of H. uvarum and Pantoea agglomerans in early-stage foods, we have added a supplemental figure showing the species-level microbial compositions corresponding to Figure 1B of the original manuscript (Figure S1), and further revised the manuscript (lines 180-186).

1. The reviewer supposes that the indigenous microbes of flies may differ from what they usually eat. In this study, the authors use banana-based food, but is it justified in terms of the natural environment of the places where those microbes were isolated? In other words, did sampled wild flies eat bananas outside the laboratory at Kyoto University?

*Drosophila* spp. inhabit human residential areas and feed on various fermented fruits and vegetables. In the areas surrounding Kyoto University, they can be found in garbage in residential dwellings as well as supermarkets. In this regard, fruits are natural food sources of wild Drosophila in the area.

Among various fruits, bananas were selected based on the following two reasons. Firstly, bananas were commonly used in previous *Drosophila* studies as a trap bait or a component of *Drosophila* food (Anagnostou et al., 2010; Stamps et al., 2012; Consuegra et al., 2020). Secondly, and rather practically, bananas can be obtained in Japan all year at a relatively low cost. Previous studies have used various fruits such as grapes (Quan and Eisen, 2018), figs (Pais et al., 2018), and raspberries (Cho and Rohlfs, 2023). However, these fruits are only available during limited seasons and are more expensive per volume than bananas. Thus, they were not practical for our study, which required large amounts of fruit-based culture media. We have included a brief explanation regarding this point in MATERIALS AND METHODS (lines 514-518).

1. In Fig. 6B, the Leu and Ile experiment, is the added amount of those amino acids appropriate in the context that they mention "...... supportive yeasts had concentrations of both leucine and isoleucine that were at least four-fold higher than those of non-supportive yeasts"?

We acknowledge that the supplementation should be carried out ideally in a quantity equivalent to the difference between the released amounts of supportive and non-supportive species. However, achieving this has been highly challenging. Previous studies determined the amount of amino acid supplementation by quantifying their concentration in the bacteriaconditioned media (Consuegra et al., 2020; Henriques et al., 2020). However, we found that quantifying the exact concentrations of the amino acids is not feasible with our yeasts. As shown in Figure 5B in the original manuscript, the amino acid contents were markedly reduced in the yeast-conditioned banana agar compared to the agar without yeasts, presumably because of the uptake by the yeasts. Thus, the amino acids released from yeast cells on the banana-agar plate are not expected to accumulate in the medium. As this reviewer pointed out, in the cell suspension supernatants of the supportive yeasts, concentrations of both leucine and isoleucine were at least four-fold higher compared to those of non-supportive yeasts (Figures 5G-H in the original submission), However, this measurement does not give the absolute amount of either amino acid available for larvae. Given these constraints, we opted for the amino acid concentrations in the holidic medium, which support larval growth under axenic conditions (Piper et al., 2014). We also showed that the supplementation of the amino acids at that concentration to the bananaagar plate was not detrimental to larval growth (Figures 6A-B in the original manuscript). These rationales have been included in the revised ‘Developmental progression with BCAA supplementation’ section in MATERIALS AND METHODS of our manuscript (lines 840-847).

1. In addition to the above, it can be included other amino acids or nutrients as control experiments.

As mentioned in our manuscript (lines 365-368), we did supplement other amino acids, lysine and asparagine, which failed to rescue the larval growth.

1. In the experiment of Fig. 2E, how about examining larval development using heat-killed LAB or yeast with live AAB? The reviewer speculates that one possibility is that AAB needs nutrients from LAB.

We did not feed larvae with heat-killed LAB and live AAB for the following reasons. LAB grows very poorly on banana agar compared to yeasts, and preparation of LAB required many banana-agar plates even when we fed live bacteria to larvae. Adding dead LAB to banana-agar tubes would require far more plates, but this preparation is impractical. Furthermore, heat-killing may not allow the investigation of the contribution of heat-unstable or volatile compounds.

As for the reviewer's suggestion regarding the addition of heat-killed yeast with AAB, heat-killed yeast itself promotes larval growth, as shown in Figures 4G and 4H in the original manuscript, so the contribution of yeast cannot be examined using this method.

Recommendations for improving the writing and presentation.1. It would be good to mention that during sample collection, other insects (other than *Drosophila* species) were not found in the food if this is true.

Insects other than *Drosophila* spp. were found in several traps in the sampling shown in Figures 1C-F. These insects, rove beetles (Staphylinidae) and sap beetles (Nitidulidae), seemed to share a niche with *Drosophila* in nature. Therefore, we believe that the contamination of these insects did not interfere with our goal of obtaining larval food samples. We added these descriptions and explanations to MATERIALS AND METHODS (lines 527531).

1. There are many different kinds of bananas. It should be mentioned the detailed information.

We had included the information on the banana in MATERIALS AND METHODS section (line 622).

1. Concerning the place of sample collection, detailed longitude, and latitude information can be provided (this is easily obtained from Google Maps). When the collection was performed should also be mentioned. This may suggest the environment of the "wild flies" they collected.

We added a table listing the dates of our collections, along with the longitude and latitude of each sampling place (Table S1A).

1. The reviewer could not find how the authors conducted heat killing of yeast.

We added the following procedure to the ‘Quantification of larval development’ section in MATERIALS AND METHODS (lines 680-688). “When feeding heat-killed yeasts to larvae, yeasts were added to the banana-agar tubes and subsequently heated as following procedures. The yeasts were revived from frozen stocks on banana-agar plates, incubated at 25°C, and then streaked on fresh agar plates. After 2-day incubation, yeast cells were scraped from the plates and suspended in PBS at the concentration of 400 mg of yeast cells in 500 µL of PBS. 125 µL of the suspensions were added to banana-agar tubes prepared as described, and after centrifugation at 3,000 x g for 5 min, the supernatants were removed. The amount of cells in each tube is ~50x compared to that when feeding live yeasts, which compensates for the reduced amount due to their inability to proliferate. The tubes were subsequently heated at 80°C for 30 min before adding germ-free larvae.”

1. The reviewer prefers that all necessary information on how to see figures be provided in figure legends. For example, an explanation of some abbreviations is missing.

We carefully re-examined the figure legends and added necessary information.

1. Many of the figures are not kind to readers, i.e., one needs to refer to the legends and main text very frequently. Adding subheadings (titles) to each figure may help.

We added subheadings to our figures to improve the comprehensibility.

**Reviewer #3 (Recommendations For The Authors):**
I have some minor questions/suggestions about the manuscript that, if addressed, may increase the clarity and quality of the work.1. Please, when referring to microbial species in the abbreviated form, use only the first letter of the genus. For example, P. agglomerans should be used, not Pa. agglomerans.

We are concerned about the potential confusion caused by using only the first letter of genera, since several genera mentioned in our work share the first letters, such as P (Pichia and Pantoea), S (Starmerella, Saccharomyces, and Saccharomycopsis), or L (Lactiplantibacillus and Leuconostoc). Therefore, we used only the unabbreviated form of the above seven genera in our revised manuscript. We have also made every effort to avoid abbreviations in our figures and tables, but found it necessary to retain two-letter abbreviations when spaces are particularly limiting.

1. In lines 294-298, how exactly was the experiment where yeasts were killed by anti-fungal agents performed? If these agents killed the yeast, how was the microbial growth on plates required to have biomass for fly inoculation obtained? Please, clarify this section.

The yeasts were grown on normal banana-agar plates before the addition onto the anti-fungal agents-containing banana agar. We added the following procedure to MATERIALS AND METHODS (lines 689-695). “When feeding yeasts on banana agar supplemented with antifungal agents, the yeasts were individually grown on normal banana agar twice before being suspended in PBS at the concentration of 400 mg of yeast cells in 500 µL of PBS. 125 µL of the suspensions was introduced onto the anti-fungal agents (10 mL/L 10% p-hydroxybenzoic acid in 70% ethanol and 6 mL/L propionic acid, following the concentration described in Kanaoka et al., 2023)-containing banana agar in 1.5 mL tubes. After centrifugation, the supernatants were removed. The amount of cells in each tube is ~50x compared to that when feeding live yeasts.”

1. In lines 557-558, please clarify how rDNA copy numbers can be calculated in this way.

Considering the results of the ITS and 16S sequencing analysis, it was highly likely that rDNAs from bananas and *Drosophila* were amplified along with microbial rDNA in this qPCR. To estimate the microbial rDNA copy number, we assumed that the proportion of microbial rDNA within the total amplification products remains consistent between the qPCR and the corresponding sequencing analysis, because the template DNA samples and amplified regions were shared between the analyses. Based on this, the copy number of microbial rDNA was estimated by multiplying the qPCR results with the microbial rDNA ratio observed in the ITS or 16S sequencing analysis of each sample. This methodology has been detailed in the MATERIALS AND METHODS section (lines 609-615).

1. In lines 609-611, how did you check for cells left from the previous day? Microscopy? Or do you mean that if there was liquid still in the sample you would not add more bacterial cultures? Please, clarify.

We observed with the naked eye from outside the tubes to determine if additional AAB should be introduced. Since we placed AAB on the banana agar in a lump, we examined whether the lumps were gone or not. We have added these procedures in MATERIALS AND METHODS (lines 671-673).

1. In Figure 2A, it is hard to differentiate between the gray tones. Please, improve this.

We have distinguished the plots for different conditions by changing the shape of the markers on the graphs.

1. In the legend of Figure 4, line 1101, I believe the panel letters are incorrect.

We have corrected the manuscript (lines 1241-1242) from “heat-killed yeasts on banana agar (H and I) or live yeasts on a nutritionally rich medium (J and K)” to “heat-killed yeasts on banana agar (G and H) or live yeasts on a nutritionally rich medium (I and J).”

1. In Figure S1, authors showed that bananas that were not inoculated still had detectable rDNA signal. Is this really because bacteria can penetrate the peel? Or could this be the “reagent microbiome”? Alternatively, could these microbes have been introduced during sample prep, such as cutting the bananas?

The detection of rDNA in bananas that were not inoculated with microbes was unlikely to be due to microbial contamination during experimental manipulation. The reviewer pointed out the possibility that the “reagent microbiome”, presumably the microbes in PBS, are detected from the uninoculated bananas. This seems to be unlikely, considering the PBS was sterilized by autoclaving before use. To ensure that no viable microbe was left in the autoclaved PBS, we applied a portion of the PBS onto a banana-agar plate and confirmed no colony was formed after incubation for a few days. DNA derived from dead microbes might be present in the PBS, but the PBS-added bananas were incubated for 4 days, so it is also unlikely that a detectable amount of DNA remained until sample collection. Furthermore, we believe that no contamination occurred during sample preparation. Banana peels were treated with 70% ethanol before removing them extremely carefully to avoid touching the fruit inside. All tools were sterilized before use. Taking all of these into account, we speculate that the microbes were already present in the bananas before peeling. We added the details of the sample preparation processes in MATERIALS AND METHODS (lines 518-521 and 540).

Other major revisions

1. We deposited our yeast genome annotation data in the DDBJ Annotated/Assembled Sequences database, and the accession numbers have been added to the ‘Data availability’ section in MATERIALS AND METHODS (lines 868-873).

2. The bacterial composition data in Figure 1B was corrected, because in the original version, the data for Place 3 and Place 4 was plotted in reverse. The original and revised plots are shown side by side in Author response image 3. We hope that the reviewers agree that this replacement of the plots does not affect our conclusion (p5, lines 117-120).

**Author response image 3. sa4fig3:** Comparison of the original and revised version of bacterial composition graph in Figure 1B. Comparison of the original (left) and revised (right) version of the graph at the bottom of Figure 1B, which shows the result of bacterial composition analysis. The color key, which is unmodified, is placed below the revised version.

1. The plot data and labels in the RNA-seq result heatmaps (Figures 3A and 4C) have been corrected. In these figures, row Z-scores of log2(TPM + 1) were to be plotted, as indicated by the key in each figure. However, in the original version, row Z-scores ofTPM was erroneously plotted. Thus, Figures 3A and 4C of the original version have been replaced with the correct plots, and the original and revised plots are shown side by side in Author response images 4A and 4B. We hope that the reviewers agree that this replacement of the plots does not affect our conclusion (p7, lines 222-226 and p9, lines 277-281).

**Author response image 4. sa4fig4:** Comparison of the original and revised version of Figures 3A and 4C. (A and B) Comparison of the original (left) and revised (right) version of Figures 3A (A) or 4C (B).

1. The keys in the original Figures 3D and 4F indicate that log2(fold change) was used to plot all data. However, when plotting the data from the previous study (Zinke et al., 2002), their “fold change value” was used. We have corrected the keys, plots, and legend of Figure 3D to reflect the different nature of the data from our RNA-seq analysis and those from microarray analysis by Zinke et al. The original and revised plots are shown side by side in Author response image 5. We hope that the reviewers agree that this replacement of the plots does not affect our conclusion (p7, lines 228230 and p9, 277-284).

**Author response image 5. sa4fig5:** Comparison of the original and revised version of Figures 3D and 4F. (A and B) Comparison of the original (left) and revised (right) version of Figures 3D (A) or 4F (B).

1. The labels in Figure S5C and S5D (Figure S4C and S4D in the original version) have been corrected (they are "Pichia kluyveri > Supportive" and "Starmerella bacillaris > Supportive" rather than "Non-support. > H. uva" and "Non-support. > K. hum").Additionally, we have reintroduced the circle indicating the number of “dme04070:Phosphatidylinositol signaling system” DEGs in Figure S5D, which was missing in Figure S4D of the original version. The original and revised figures are shown in Author response image 6.

**Author response image 6. sa4fig6:** Comparison of the original and revised version of Figures S5C and S5D. (A and B) Comparison of the original (left) and revised (right) versions of Figures S5C (A) or S5D (B). The original figures corresponding to the aforementioned figures were Figures S4C and S4D, respectively.

1. The "Fermentation stage" column in Table 1, which indicated whether each microbe was considered an early-stage microbe or a late-stage microbe, has been removed to avoid confusion. This is because some of the microbes (Hanseniaspora uvarum, Pichia kluyveri, and Pantoea agglomerans) were employed in both of the feeding experiments using the microbes detected from the early-stage foods (Figures 2A, 2B, S2A, and S2B) and those from the late-stage foods (Figures 2C, 2D, S2C, and S2D).

2. The leftmost column in Table S7 has been edited to indicate species names rather than “Sample IDs,” because the IDs were not used in anywhere else in the paper.

Reference

Chandler, J. A., Lang, J., Bhatnagar, S., Eisen, J. A. and Kopp, A. (2011). Bacterial communities of diverse *Drosophila* species: Ecological context of a host-microbe model system. PLoS Genetics 7, e1002272.

Chandler, J. A., Eisen, J. A. and Kopp, A. (2012). Yeast communities of diverse *Drosophila* species: Comparison of two symbiont groups in the same hosts. Applied and Environmental Microbiology 78, 7327–7336.

Cho, H. and Rohlfs, M. (2023). Transmission of beneficial yeasts accompanies offspring production in *Drosophila*—An initial evolutionary stage of insect maternal care through manipulation of microbial load? Ecology and Evolution 13, e10184.

Consuegra, J., Grenier, T., Akherraz, H., Rahioui, I., Gervais, H., da Silva, P. and Leulier, F. (2020). Metabolic Cooperation among Commensal Bacteria Supports *Drosophila* Juvenile Growth under Nutritional Stress. iScience 23, 101232.

Dodge, R., Jones, E. W., Zhu, H., Obadia, B., Martinez, D. J., Wang, C., Aranda-Díaz, A., Aumiller, K., Liu, Z., Voltolini, M., et al. (2023). A symbiotic physical niche in *Drosophila melanogaster* regulates stable association of a multi-species gut microbiota. Nat Commun 14, 1557.

Erkosar, B., Storelli, G., Mitchell, M., Bozonnet, L., Bozonnet, N. and Leulier, F. (2015). Pathogen Virulence Impedes Mutualist-Mediated Enhancement of Host Juvenile Growth via Inhibition of Protein Digestion. Cell Host & Microbe 18, 445–455.

Hanson, M. A. and Lemaitre, B. (2020). New insights on *Drosophila* antimicrobial peptide function in host defense and beyond. Current Opinion in Immunology 62, 22–30.

Henriques, S. F., Dhakan, D. B., Serra, L., Francisco, A. P., Carvalho-Santos, Z., Baltazar, C., Elias, A. P., Anjos, M., Zhang, T., Maddocks, O. D. K., et al. (2020). Metabolic cross-feeding in imbalanced diets allows gut microbes to improve reproduction and alter host behaviour. Nat Commun 11, 4236.

Oka, M., Hashimoto, K., Yamaguchi, Y., Saitoh, S., Sugiura, Y., Motoi, Y., Honda, K., Kikko, Y., Ohata, S., Suematsu, M., et al. (2017). Arl8b is required for lysosomal degradation of maternal proteins in the visceral yolk sac endoderm of mouse embryos. Journal of Cell Science jcs.200519.

Pais, I. S., Valente, R. S., Sporniak, M. and Teixeira, L. (2018). *Drosophila melanogaster* establishes a species-specific mutualistic interaction with stable gut-colonizing bacteria. PLOS Biology 16, e2005710.

Piper, M. D. W., Blanc, E., Leitão-Gonçalves, R., Yang, M., He, X., Linford, N. J., Hoddinott, M. P., Hopfen, C., Soultoukis, G. A., Niemeyer, C., et al. (2014). A holidic medium for *Drosophila melanogaster*. Nature Methods 11, 100–105.

Piper, M. D. W., Soultoukis, G. A., Blanc, E., Mesaros, A., Herbert, S. L., Juricic, P., He, X., Atanassov, I., Salmonowicz, H., Yang, M., et al. (2017). Matching Dietary Amino Acid Balance to the In Silico-Translated Exome Optimizes Growth and Reproduction without Cost to Lifespan. Cell Metab 25, 610–621.

Quan, A. S. and Eisen, M. B. (2018). The ecology of the *Drosophila*-yeast mutualism in wineries. PLOS ONE 13, e0196440.

Solomon, G. M., Dodangoda, H., McCarthy-Walker, T. T., Ntim-Gyakari, R. R. and Newell, P. D. (2019). The microbiota of *Drosophila* suzukii influences the larval development of *Drosophila melanogaster*. PeerJ 7, e8097.

Zinke, I., Schütz, C. S., Katzenberger, J. D., Bauer, M. and Pankratz, M. J. (2002). Nutrient control of gene expression in *Drosophila*: microarray analysis of starvation and sugar-dependent response. The EMBO Journal 21, 6162–6173.